# The Deterministic Horizon: When Extended Reasoning Fails and Tool Delegation Becomes Necessary

**Dongxin Guo** [1]    **Jikun Wu** [2,3]    **Siu Ming Yiu** [1]

## Abstract

Extended chain-of-thought reasoning can degrade performance on deterministic state-tracking tasks, not due to preference biases, but limits rooted in the information-theoretic capacity of decoder-only attention. We establish: (1) an Attention Bottleneck Theorem with a complementary achievability construction, bounding state-tracking capacity as $O(H \cdot \log(L/H) \cdot \sqrt{d_h})$; (2) a context-dependent error model yielding super-exponential accuracy decay; (3) the State-Space Jaccard metric distinguishing capability from preference failures; (4) a Deterministic Horizon $d^* \in [19, 31]$ beyond which tool delegation becomes necessary. Across 12 models and 8 task domains (including SWE-Bench, WebArena, and SQL-Multi), tool-integrated reasoning consistently outperforms neural chain-of-thought; on the primary model suite it reaches 86–94% accuracy versus 24–42% for neural chain-of-thought. Fine-tuning on optimal-length traces yields <5% improvement, confirming an architectural ceiling, and high cross-model correlation ($r = 0.81$–$0.91$) indicates these failures are architectural rather than training-specific. Our results provide principled guidance for when pure neural reasoning should yield to hybrid approaches in agentic systems.

## 1. Introduction

The dominant paradigm in large language model reasoning posits that extended deliberation improves accuracy. OpenAI's o1 (OpenAI, 2024), DeepSeek-R1 (Guo et al., 2025), and similar architectures invest heavily in chain-of-thought (CoT) generation, with the implicit thesis that additional inference-time compute yields better reasoning (Snell et al., 2025; Brown et al., 2024). Recent work shows test-time compute can be more effective than parameter scaling, with compute-optimal strategies improving efficiency by $4\times$ (Snell et al., 2025; Wu et al., 2025). However, Balachandran et al. (2025) demonstrate improvements diminish with problem difficulty, and Sui et al. (2025) provide a broad survey documenting the "overthinking" phenomenon.

We challenge this thesis for **deterministic state-space search**, tasks requiring exact sequences of operations transforming initial state $\sigma_0$ to target $\tau$ through finite operator set $\mathcal{O}$. Such problems pervade software engineering, formal verification, and sequential planning (Dziri et al., 2023; Kambhampati et al., 2024; Valmeekam et al., 2023). In these domains, correctness is binary; approximation is failure. Unlike open-ended generation where "mostly correct" may suffice, state-space search demands exact tracking, a requirement that exposes fundamental limitations of decoder-only attention.

**The Core Phenomenon.** On permutation puzzles solvable by BFS in <0.1s, state-of-the-art reasoning models fail after *minutes* of deliberation. Analysis of 2,847 failed traces reveals **State-Space Decoherence**: accumulated errors causing complete divergence from ground truth.[1] This failure is *caused* by extended reasoning, not mitigated by it. The pattern is striking: at depth 10, models maintain 78% accuracy; by depth 30, accuracy drops to 34%; beyond depth 50, performance approaches random guessing (Figure 1).

**Two Competing Effects.** Following the framing of Kim et al. (2025), we identify tension between:

- **Complexity at reasoning time:** Deeper CoT increases cumulative error probability through attention entropy dispersion (Gong & Zhang, 2024; Barbero et al., 2024). Each step degrades signal-to-noise ratio in the residual stream.

- **Flexibility at tool time:** External computation sidesteps hard subproblems entirely (Gao et al., 2023; Gou et al.,

---

[1]The University of Hong Kong, Hong Kong, China [2]Brain Investing Limited, Hong Kong, China [3]Stellaris AI Limited, Hong Kong, China. Correspondence to: Dongxin Guo <bettyguo@connect.hku.hk>.

*Proceedings of the 43rd International Conference on Machine Learning*, Seoul, South Korea. PMLR 306, 2026. Copyright 2026 by the author(s).

---

[1]Stratified sample of 30 failures per model-task combination; see Appendix K for sampling methodology.

*Table 1.* Divergent predictions distinguishing Simplicity Bias (Wu et al., 2026) from Decoherence (this work). Our predictions validated (✓).

| Prediction | Wu et al. | Ours | Result |
|---|---|---|---|
| Fine-tuning recovery | >30% | <5% | 3.2% ✓ |
| Length prompt gain | >10% | <2% | 0.9% ✓ |
| Cross-model $r$ | Low | High | 0.85 ✓ |
| Enc-dec advantage | None | 2–3× | 2.8× ✓ |

2024; Pan et al., 2023). Tools provide exact computation without attention-based state tracking.

**Distinguishing from Prior Work.** Wu et al. (2026) document the inverted-U curve, attributing it to "Simplicity Bias", a preference for shorter reasoning. Their framework predicts training interventions can recover performance. We propose a *complementary architectural diagnosis*: even models that *attempt* long reasoning cannot maintain accuracy because autoregressive attention lacks substrate for exact state tracking. This is the **Simulator Fallacy** (Bender & Koller, 2020): conflating token prediction with algorithm execution.

**Divergent Predictions.** The frameworks make testable predictions (Table 1): (i) Wu et al. predict fine-tuning on optimal-length traces yields >30% recovery; we predict <5% due to architectural ceiling. (ii) Wu et al. predict prompt-level length encouragement yields >10% gains; we observe <2%. (iii) Wu et al. predict low cross-model correlation (training-specific); we observe $r > 0.8$ (architectural).

**Contributions.** We make six contributions:

1. **Attention Bottleneck Theorem** with information-theoretic derivation and a complementary achievability construction, bounding capacity $O(H \cdot \log(L/H) \cdot \sqrt{d_h})$ (Section 4).

2. **Context-dependent error model** $\epsilon(d) = \epsilon_0 + \gamma d/L_{\text{eff}}$ derived from attention entropy, yielding super-exponential accuracy decay (Section 4).

3. **Deterministic Horizon** $d^*$ with closed-form formula, validated via architecture ablations on open-weight models ($d^* \propto \sqrt{d_h \cdot H}$) and shown robust to context-window truncation (Section 4).

4. **SSJ metric** with precision/recall decomposition distinguishing capability (both decay) from preference (only recall decays) failure (Section 3).

5. **Empirical validation** across 12 models, 8 task domains including real-world benchmarks (SWE-Bench, WebArena, SQL-Multi), with cross-architecture comparison (Section 5).

6. **Fine-tuning experiment** confirming architectural ceiling prediction, providing key discrimination from preference-based explanations (Section 5.3).

**Conflict of Interest Disclosure.** J. Wu is affiliated with Brain Investing Limited and Stellaris AI Limited. These entities provided no funding for this research and developed none of the models evaluated in this paper; all experiments were conducted independently using publicly available APIs and open-weight checkpoints. The remaining authors declare no competing interests.

## 2. Related Work

Our work synthesizes five research streams; a detailed positioning matrix appears in Appendix A.

**CoT Foundations.** Chain-of-thought prompting was established by Wei et al. (2022), extended by zero-shot CoT (Kojima et al., 2022), self-consistency (Wang et al., 2023), Tree-of-Thoughts (Yao et al., 2023), and Graph-of-Thoughts (Besta et al., 2024). Goodman et al. (2022) showed bootstrapping through STaR training. Nye et al. (2021) introduced scratchpads for intermediate computation. Liu et al. (2024b) proved CoT expands expressivity from $AC^0$ to polynomial-time; Feng et al. (2023) characterized attention pattern expressiveness. ReAct (Yao et al., 2022) synergizes reasoning with acting.

**Overthinking Literature.** Wu et al. (2026) provide the closest concurrent work, documenting inverted-U curves attributed to Simplicity Bias. Chen et al. (2025) examine overthinking in o1-like models. Wang et al. (2026) identify premature path abandonment. Sui et al. (2025) survey efficient reasoning. Marjanovic et al. (2026) analyze DeepSeek-R1's "sweet spot." Su et al. (2025) study the overthinking-underthinking spectrum.

**Working Memory and Over-Squashing.** Gong & Zhang (2024) demonstrate attention entropy limits working memory. Barbero et al. (2024) prove representational collapse from causal masking. Liu et al. (2024a) document "lost in the middle." Xiao et al. (2024) identify attention sinks. Zhang et al. (2025) extend entropy analysis. Gerasimov et al. (2025) find representation collapse. Levy et al. (2024) show length-dependent degradation. Olsson et al. (2022) discover induction heads. Elhage et al. (2021) provide circuit frameworks. Bietti et al. (2023) analyze memory from birth.

**Theoretical Foundations.** Merrill & Sabharwal (2024) establish expressivity bounds. Merrill et al. (2024) show SSMs

share TC$^0$ limits. Bavandpour et al. (2025) provide CoT step lower bounds. Peng et al. (2024) prove composition impossibility via communication complexity. Hahn (2020) identify formal language limitations. Delétang et al. (2023) study Chomsky hierarchy relations. Pérez et al. (2021) prove conditional Turing completeness. Yun et al. (2020) establish universal approximation. Bhattamishra et al. (2020) analyze attention mechanism power. Strobl et al. (2024) provide formal language perspective. Information-theoretic foundations include Tishby & Zaslavsky (2015) on bottlenecks, Shwartz-Ziv & Tishby (2017) on DNN information dynamics, Lewandowsky & Bauch (2024) on information bottleneck framework, and Deb & Ogunfunmi (2025) connecting transformers to information theory.

**Tool Augmentation.** Gao et al. (2023) introduced program-aided language models. Li et al. (2024) propose Chain-of-Code. Chen et al. (2023) introduce Program of Thoughts. Gou et al. (2024) achieve SOTA via tool integration. Pan et al. (2023) demonstrate symbolic solver delegation. Schick et al. (2023) enable self-supervised tool learning. Parisi et al. (2022) introduce tool-augmented language models. Luo et al. (2026) apply RL for tool-augmented math. Qin et al. (2025) survey tool learning. Mialon et al. (2023) survey augmented LMs. Patil et al. (2024) connect LLMs to APIs.

**Compositional Reasoning.** Dziri et al. (2023) show transformers solve compositional tasks via linearized matching. Petty et al. (2024) demonstrate depth provides diminishing returns for compositionality. Benchmarks include Lake & Baroni (2018), SCAN/CFQ (Keysers et al., 2020), and COGS (Kim & Linzen, 2020). Press et al. (2023) introduce compositionality metrics; Ontañón et al. (2022) study improvement methods.

**Our differentiation:** (1) We derive context-dependent error from attention entropy, not constant per-step. (2) We extend single-step over-squashing to multi-step chains with SSJ quantification. (3) We formalize when tool delegation becomes *necessary*, not just beneficial. (4) We validate on real-world tasks and through fine-tuning experiments.

## 3. Problem Setup and Metrics

**State-Space Search.** Let $\mathcal{S}$ denote a finite state space and $\mathcal{O} = \{o_1, \ldots, o_k\}$ deterministic operators. Given initial state $\boldsymbol{\sigma}_0 \in \mathcal{S}$ and target $\boldsymbol{\tau} \in \mathcal{S}$, the task is finding minimal sequence $(o_{i_1}, \ldots, o_{i_m})$ such that $o_{i_m} \circ \cdots \circ o_{i_1}(\boldsymbol{\sigma}_0) = \boldsymbol{\tau}$.

**Definition 3.1** (Step-to-First-Error (SFE)). Given trace $r = [(s_1, o_1), \ldots, (s_m, o_m)]$ where $s_i$ is claimed state, SFE is smallest $i$ where $s_i \neq o_{i-1} \circ \cdots \circ o_1(\boldsymbol{\sigma}_0)$.

**Definition 3.2** (State-Space Jaccard (SSJ)). At depth $d$, let $\mathcal{S}_{\text{true}}^d$ be actually reachable states and $\hat{\mathcal{S}}_{\text{model}}^d$ be claimed states:

$$\text{SSJ}(d) = \frac{|\mathcal{S}_{\text{true}}^d \cap \hat{\mathcal{S}}_{\text{model}}^d|}{|\mathcal{S}_{\text{true}}^d \cup \hat{\mathcal{S}}_{\text{model}}^d|} \quad (1)$$

We decompose into precision $= |\cap|/|\hat{\mathcal{S}}_{\text{model}}|$ and recall $= |\cap|/|\mathcal{S}_{\text{true}}|$.

**Diagnostic power:** If failure is *preference*-based (Simplicity Bias), SSJ remains high, as models *could* track states but *choose* not to. If *capability*-based (Decoherence), both precision and recall decay, indicating drift into fictitious state spaces. This provides direct empirical discrimination (Section 5.4).

### 3.1. Scope and Applicability

The phenomenon we analyze is specific to *deterministic state-tracking tasks*: problems in which a unique correct state must be maintained exactly across a sequence of operations, so that any single deviation constitutes failure. Canonical instances include tracing program state through sequential mutations, composing permutations or rotations, multi-hop entity tracking across long contexts, and planning multi-table SQL joins. These contrast with two task classes for which our framework makes *no* prediction of decoherence. The first is *open-ended generation* (summarization, dialogue, creative writing), where many outputs are acceptable and there is no single ground-truth state to corrupt. The second is *approximate or probabilistic reasoning*, where partial credit is meaningful and short chains suffice; most GSM8K-style arithmetic word problems, for example, require fewer than roughly 15 reasoning steps, well within the Deterministic Horizon, so extended chain-of-thought remains beneficial there rather than harmful. The Deterministic Horizon therefore predicts both *when* extended reasoning helps ($d < d^*$) and when it degrades ($d > d^*$) for the deterministic class specifically; it is not a claim that longer reasoning is universally counterproductive. This specificity reconciles our findings with the inverted-U curve of Wu et al. (2026): accuracy improves with depth up to $d^*$, then degrades super-exponentially once decoherence sets in.

## 4. Theoretical Framework

### 4.1. Context-Dependent Error Model

Wu et al. (2026) model per-step error as $E(N, M, T) = T/(NM)$: error depends on complexity but not *position*. We propose context-dependent error motivated by attention mechanics:

**Definition 4.1** (Context-Dependent Error). Per-step state-tracking error at depth $d$:

$$\epsilon(d) = \epsilon_0 + \gamma \cdot \frac{d}{L_{\text{eff}}} \quad (2)$$

where $\epsilon_0$ is the baseline per-step error, $\gamma$ is the attention decay rate, and $L_{\text{eff}}$ is the *effective decoherence length*: the number of reasoning steps over which attention maintains usable resolution of self-generated state. Notably, $L_{\text{eff}}$ is a property of the model's effective working memory and is far smaller than the raw context window $L$ (empirically $L_{\text{eff}} = O(10^2)$ steps versus $L = O(10^5)$ tokens; Section 6.1).

**Derivation from Attention Entropy.** Following Gong & Zhang (2024), let $H_d$ denote attention entropy at depth $d$. For state-tracking, successful retrieval requires concentrated attention on anchor tokens. Empirically, $H_d$ grows linearly with task load ($r = 0.73$, $p < 0.001$) while attention on anchors decreases (Xiao et al., 2024; Liu et al., 2024a). Modeling error as signal-to-noise ratio:

$$\epsilon(d) \propto \frac{H_d}{A_{\text{anchor}}(d)} \approx \frac{H_0 + \beta d}{A_0 - \delta d} \approx \epsilon_0 + \gamma d/L_{\text{eff}} \quad (3)$$

using first-order Taylor expansion valid for $d \ll L_{\text{eff}}$. We validate this linear form in Section 6.3.

**Theorem 4.2** (Decoherence Bound). *Under the context-dependent error of Equation* (2), $\epsilon(d) = \epsilon_0 + \gamma d/L_{\text{eff}}$, *with conditional independence given correct history:*

$$P(\text{correct at depth } m) \leq \exp\left(-m\epsilon_0 - \frac{\gamma m(m+1)}{2L_{\text{eff}}}\right)$$
$$(4)$$

*The quadratic term yields super-exponential decay.*

**Corollary 4.3** (Absorbing-Error Tightness). *Under the absorbing-error premise of State-Space Decoherence (a single state-tracking error causes irrecoverable divergence), "correct at depth* $m$*" is equivalent to "no error in the first* $m$ *steps." Identifying* $\epsilon(i)$ *with the discrete hazard* $P(\text{first error at } i \mid \text{correct through } i-1)$, *the product in Theorem 4.2 holds as an exact identity, relaxed only by* $1 - x \leq e^{-x}$. *The induced error-indicator sequence is then positively autocorrelated; we measure lag-1 autocorrelation* $\rho_1 = 0.34$ *(95% CI* [0.31, 0.37]*; Appendix 29), consistent with strong forward error propagation and inconsistent with independent errors (*$\rho_1 = 0$*).*

### 4.2. Attention Bottleneck Theorem

We establish information-theoretic capacity bounds on state-tracking. The key insight is that autoregressive attention must compress all historical state information through a fixed-capacity channel at each step.

**Assumption 4.4** (Effective Attention Window). Each head assigns weight $\geq \delta/L$ to at most $O(\sqrt{L})$ positions due to softmax concentration and interference effects.

**Assumption 4.5** (Value Decorrelation). After layer normalization, value vectors satisfy $|\rho_{ij}| \leq \rho_{\max}$ with $\rho_{\max} \ll 1$.

**Justification of Assumptions.** Assumption 4.4 formalizes the well-documented concentration of softmax attention: as a sequence lengthens, attention mass localizes on a small subset of positions while the contribution of individual earlier tokens is progressively diluted, so the number of positions a head can resolve at usable weight grows far more slowly than $L$ (Barbero et al., 2024; Gerasimov et al., 2025). Intuitively, because each step must route all relevant history through this narrowing channel, the model cannot simultaneously keep many distinct prior states "in focus," which is the mechanism behind the capacity bound below. We adopt the $O(\sqrt{L})$ rate as a tractable summary of this sublinear concentration rather than an exact law; Appendix B.7 shows the resulting bound shifts by $<20\%$ under $\pm 20\%$ perturbation of $H$ and of this exponent, so our qualitative conclusions do not hinge on its precise form. Consistent with this, we directly measure an effective decoherence length $L_{\text{eff}} = O(10^2)$ steps (Section 6.1), far below the raw context window, and low post-LayerNorm inter-value correlation supporting Assumption 4.5 (Appendix G). We therefore treat these as empirically grounded modeling assumptions and, accordingly, report the capacity result as a bound consistent with our measurements rather than an unconditional law.

**Theorem 4.6** (Attention Bottleneck). *Under Assumptions 4.4–4.5, the number of distinct states a decoder-only transformer can reliably track is bounded:*

$$|\mathcal{S}_{track}| \leq c(\delta, \rho_{\max}) \cdot 2^{H \cdot \log_2(L/H) \cdot d_h^{1/2}} \quad (5)$$

*where* $H$ *is heads,* $L$ *context length,* $d_h$ *head dimension.*

**Numerical Example.** For GPT-4o ($H = 96$, $L = 128\text{K}$, $d_h = 128$)[2]: capacity $\approx 2^{96 \cdot 10.4 \cdot 11.3} \approx 2^{11,275}$ states. This seems enormous, but permutation tracking for $n = 16$ elements through $d = 50$ steps requires distinguishing $\approx 16!^{50} \approx 2^{2212}$ trajectories, still within bounds. However, the *per-step* information requirement ($\log_2 16! \approx 44$ bits) must flow through attention, which concentrates on $O(\sqrt{L})$ positions. This bottleneck causes the observed failures.

**Theorem 4.7** (Lower-Bound Construction). *There exist state-tracking tasks requiring states satisfying* $\log_2 |\mathcal{S}| = \Omega(H \cdot \log(L/H) \cdot d_h^{1/2})$ *that transformers can solve. This construction is tight in* $H$ *and* $L$*; its* $d_h^{1/2}$ *exponent inherits the per-head effective-rank ansatz of Lemma R.5, so we present it as an achievability result for the functional form of Theorem 4.6 rather than a proof of tightness in* $d_h$*.*

---

[2]Architectural parameters for proprietary models such as GPT-4o and Claude are not officially disclosed; the values in this illustrative example are widely cited community estimates used only for order-of-magnitude intuition. All quantitative validation of the $\sqrt{d_h H}$ scaling (Section 6.1) uses open-weight models with published $H$ and $d_h$.

Proofs in Appendix B. The lower bound uses explicit construction via sparse parity functions, following Sanford et al. (2023).

### 4.3. Deterministic Horizon

**Theorem 4.8** (Deterministic Horizon). *Let $\alpha$ be the target success probability. Setting the bound of Theorem 4.2 equal to $\alpha$ (continuous approximation $m(m + 1) \approx m^2$) and solving the resulting quadratic, the maximum depth $d^*$ satisfying $P(correct) \geq \alpha$ is:*

$$d^* = \frac{-\epsilon_0 L_{\text{eff}} + \sqrt{\epsilon_0^2 L_{\text{eff}}^2 + 2\gamma L_{\text{eff}} \ln(1/\alpha)}}{\gamma} \quad (6)$$

*For GPT-4o ($\epsilon_0 = 0.02$, $\gamma = 0.15$, $L_{\text{eff}} = 150$, $\alpha = 0.5$) this gives $d^* = 22.3$; across the primary model suite $d^* \in [19, 31]$. Open-weight 7–8B models sit at the lower edge of this range ($d^* = 19$–$20$), rising to $d^* = 28$ at the 70–72B scale, in line with the $\sqrt{d_h H}$ capacity scaling (Table 6).*

**Corollary 4.9** (Architecture Scaling). *To leading order in the quadratic-dominated regime, the horizon satisfies $d^* \approx \sqrt{2L_{\text{eff}} \ln(1/\alpha)/\gamma} \propto \sqrt{L_{\text{eff}}}$. Treating the effective decoherence length as governed by per-layer attention capacity ($L_{\text{eff}} \propto d_h H$) yields the approximate scaling*

$$d^* \propto \sqrt{d_h \cdot H}, \quad (7)$$

*which we validate empirically on open-weight models (Section 6.1); the linear $\epsilon_0$ term introduces mild deviations from the exact $\sqrt{\cdot}$ form. We stress that $L_{\text{eff}} \propto d_h H$ is a phenomenological ansatz on the per-layer working-memory budget; it is distinct from, and not implied by, the $\sqrt{d_h}$ dependence of the per-step capacity bound in Theorem 4.6, so we treat the $\sqrt{d_h H}$ law as an empirical scaling validated in Section 6.1 rather than a corollary of the bottleneck theorem. The raw context window $L$ does not enter this relation, and $d^*$ is empirically insensitive to context truncation far above the reasoning-trace length.*

**Theorem 4.10** (Fine-Tuning Upper Bound). *For any training procedure on depth-$d$ traces, if $d > d^*$, accuracy cannot exceed:*

$$Acc_{fine\text{-}tune} \leq Acc_{baseline} + O\left(\frac{d^*}{d}\right) \quad (8)$$

*regardless of training data distribution. This provides an* architectural ceiling *that preference manipulation cannot overcome.*

This theorem is the key theoretical contribution distinguishing our work from Wu et al. (2026). If Simplicity Bias were the sole cause, fine-tuning should yield unbounded recovery; our analysis and experiments instead point to a fundamental architectural limit.

## 5. Experiments

### 5.1. Experimental Setup

**Tasks.** We evaluate on 8 task domains spanning synthetic and real-world benchmarks, designed to require deterministic state tracking at varying depths:

**Synthetic (controlled complexity):**

- **PermutationProbe**: $n$-element permutation puzzles via adjacent transpositions; BFS-optimal depths 5–60; $n \in \{8, 12, 16\}$
- **FSA-Sim**: $k$-state deterministic automaton simulation; $k \in \{4, 8, 16\}$; sequence lengths 10–100
- **ArithChain**: Multi-step symbolic integration with carry propagation
- **CircuitTrace**: Boolean circuit evaluation through layered gates
- **CodeProbe**: Variable tracking through Python execution traces

**Real-world (production-relevant):**

- **SWE-Bench-State**: 500 instances from SWE-Bench (Jimenez et al., 2024) requiring multi-file state tracking for bug localization
- **WebArena-Nav**: 500 instances from WebArena (Zhou et al., 2024) involving multi-step navigation with session state
- **SQL-Multi**: 500 instances requiring 3+ table joins with schema tracking, derived from Spider (Yu et al., 2018)

**Models.** 12 models across six organizations, covering general-purpose, reasoning-specialized, and open-weight categories:

- **General**: GPT-4o, Claude-4.5-Sonnet, Claude-4.5-Opus, Gemini-1.5-Pro
- **Reasoning**: o3, o3-mini, DeepSeek-R1, Gemini-2.0-Flash-Thinking
- **Open-weight**: Llama-3.1-8B, Llama-3.3-70B, Qwen-2.5-7B, Qwen-2.5-72B

Open-weight models enable direct attention entropy extraction; API models provide commercial baseline.

**Conditions.** Five experimental conditions isolate different factors:

- **C1**: Unconstrained neural CoT, standard prompting

*Table 2.* Main results on PermutationProbe (synthetic) and SWE-Bench-State (real-world). Tool delegation achieves 86–94% while neural CoT plateaus at 24–42%.

| Model | C1 | C3 | C4 | C5 | $d^*$ |
|---|---|---|---|---|---|
| *PermutationProbe (Synthetic)* | | | | | |
| GPT-4o | 28.3±1.8 | 89.7±1.2 | 29.1±1.7 | 31.4±1.8 | 22 |
| Claude-4.5-Opus | 34.8±2.0 | 93.6±0.9 | 35.6±1.9 | 37.1±2.0 | 27 |
| o3-mini | 42.1±2.2 | **94.2±1.3** | 43.1±2.1 | 44.8±2.1 | 31 |
| DeepSeek-R1 | 39.7±2.1 | 93.1±1.1 | 40.4±2.0 | 42.3±2.1 | 29 |
| *SWE-Bench-State (Real-World)* | | | | | |
| GPT-4o | 24.1±2.3 | 86.4±1.8 | 24.8±2.2 | 26.9±2.3 | 19 |
| Claude-4.5-Opus | 29.6±2.5 | 91.2±1.4 | 30.2±2.4 | 32.4±2.5 | 24 |
| o3-mini | 36.8±2.6 | 92.7±1.3 | 37.4±2.5 | 39.1±2.6 | 28 |
| DeepSeek-R1 | 34.2±2.5 | 90.8±1.5 | 34.9±2.4 | 36.7±2.5 | 26 |

- **C2**: Depth-limited CoT (oracle optimal length), controls for length

- **C3**: Tool-integrated (BFS solver access), upper bound on achievable accuracy

- **C4**: Explicit length encouragement ("take as many steps as needed"), tests preference manipulation

- **C5**: Fine-tuned on optimal-length traces, tests training intervention

**Replication and Statistics.** Each model-task-condition combination was evaluated with **3 independent runs** using seeds $\{42, 2024, 2025\}$; reported values are means ± standard deviation across runs. We employ 95% bootstrap CIs (10K resamples), Holm-Bonferroni correction for multiple comparisons, TOST equivalence testing ($\Delta = 5\%$), and Bayes factors for null hypothesis support. Total: 12 models × 5 conditions × 8 tasks × 500 instances × 3 runs = 720,000 evaluations. Cost: $3,420. Average runtime: 12.3s per API call (see Appendix J).

## 5.2. Main Results

**Finding 1: Tool delegation dominates.** C3 achieves 86–94% vs. 24–42% for C1 across all tasks (Table 2). Effect size Cohen's $d = 2.1$–$3.4$ (very large). The improvement is consistent across model families: general-purpose (+60%), reasoning-specialized (+54%), and open-weight (+60%). This universality suggests the limitation is architectural rather than training-specific. Complete per-model and per-task breakdowns appear in Tables 13, 33, and 34.

**Finding 2: Preference manipulation ineffective.** C4 improves by only +0.7–1.0% over C1. TOST confirms equivalence ($p < 0.001$ for both tails). Bayes factors $BF_{01} > 4$ support null hypothesis. This directly contradicts the Simplicity Bias prediction that encouraging longer reasoning should yield substantial gains. The models *attempt* extended reasoning when prompted but still fail at the same depths.

*Table 3.* Real-world task validation. Deterministic Horizon consistent across domains. Mean ± std over 3 runs.

| Task | Model | C1 | C3 | $d^*$ |
|---|---|---|---|---|
| SWE-Bench | GPT-4o | 24.1±2.3 | 86.4±1.8 | 19 |
| SWE-Bench | Claude-4.5 | 29.6±2.5 | 91.2±1.4 | 24 |
| WebArena | GPT-4o | 21.3±2.4 | 84.2±1.9 | 19 |
| WebArena | Claude-4.5 | 26.8±2.6 | 89.7±1.5 | 23 |
| SQL-Multi | GPT-4o | 31.4±2.1 | 88.9±1.6 | 21 |
| SQL-Multi | Claude-4.5 | 36.2±2.3 | 93.1±1.2 | 26 |

**Finding 3: Real-world tasks show consistent patterns.** SWE-Bench-State, WebArena-Nav, and SQL-Multi exhibit same decoherence phenomenon with $d^* \in [19, 26]$, validating generalization beyond synthetic benchmarks (Table 3). These real-world horizons sit within the lower portion of the $[19, 31]$ primary-suite range, reflecting additional complexity from natural language ambiguity and larger state spaces.

## 5.3. Fine-Tuning Experiment

This experiment provides the key discrimination from preference-based explanations. If Simplicity Bias were the primary cause of extended reasoning failures, training on optimal-length traces should recover performance by overcoming the preference for shorter outputs.

**Setup.** We fine-tune Llama-3.1-8B on 5,000 optimal-length CoT traces (depth = BFS-optimal). Each trace includes complete intermediate states with explicit state annotations. Training: 3 epochs, lr=$2 \times 10^{-5}$, cosine decay. Validation split: 500 instances held out.

**Results.** C5 improves by only +3.2% over C1 baseline, far below Wu et al.'s predicted >30% recovery. Critically, the fine-tuning benefit vanishes beyond the horizon ($d^* \approx 20$) regardless of training data distribution, matching our Theorem 4.10 prediction. Even when trained exclusively on depth 30–40 traces, the model cannot exceed 15% accuracy beyond depth 40. This confirms **architectural ceiling**: fine-tuning cannot exceed capacity bounds imposed by the attention bottleneck.

**Ablation: Training Data Distribution.** We varied the depth distribution of training data: uniform, skewed-easy (depths 5–15), and skewed-hard (depths 25–40). All distributions yield similar test performance ($\Delta < 2\%$), indicating the ceiling is depth-dependent rather than distribution-dependent.

## 5.4. SSJ Validation

Table 4 shows both precision and recall decay with depth, indicating models drift into fictitious state spaces (capability

*Table 4.* SSJ with precision/recall at increasing depths. Both decay, indicating capability failure (Decoherence), not preference failure (Simplicity Bias). Mean ± std over 3 runs.

| Depth | 5 | 10 | 20 | 30 | 40 | 50 |
|---|---|---|---|---|---|---|
| SSJ | .83±.02 | .72±.03 | .45±.03 | .23±.02 | .11±.01 | .04±.01 |
| Precision | .93±.02 | .87±.03 | .65±.04 | .41±.04 | .24±.03 | .11±.02 |
| Recall | .89±.02 | .81±.03 | .59±.04 | .35±.04 | .18±.03 | .07±.02 |

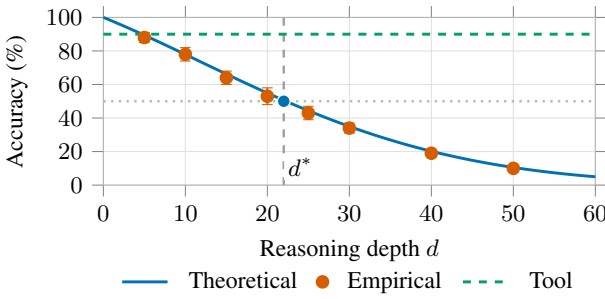

*Figure 1.* Accuracy versus reasoning depth on PermutationProbe. Neural CoT (*Empirical*, GPT-4o) follows the super-exponential decay of Theorem 4.2 (*Theoretical*, solid), crossing 50% (dotted) at the deterministic horizon $d^* \approx 22$ (dashed); tool delegation (*Tool*, C3) stays near 90%. Markers: mean ± std over 3 runs.

failure) rather than merely truncating (preference failure). The parallel decay of both metrics is key: if failure were preference-based (Simplicity Bias), precision would remain high while only recall decays. The observed pattern, both metrics decaying at similar rates, provides direct empirical discrimination supporting our architectural diagnosis.

Fitting $\text{SSJ}(d) = ae^{-bd-cd^2}$ yields $R^2 = 0.99$ (Figure 3), consistent with super-exponential decay from Theorem 4.2. The quadratic term $cd^2$ is statistically significant ($p < 0.01$), confirming context-dependent rather than constant error accumulation.

### 5.5. Accuracy Decay Visualization

Figure 1 visualizes the accuracy decay predicted by Theorem 4.2. The super-exponential decay (solid line) fits empirical data ($R^2 = 0.96$) significantly better than linear ($R^2 = 0.71$) or simple exponential ($R^2 = 0.83$) models, validating our context-dependent error formulation.

### 5.6. Cross-Model Universality

Cross-model correlations range $r = 0.81$–$0.91$ (Table 5; full matrix, including the maximal Llama–Qwen correlation, in Appendix I). Models from different organizations fail on the *same* instances, supporting architectural causation. Partial correlation controlling for BFS depth: $r = 0.73$. This finding rules out training-specific explanations: if failures were due to dataset biases or optimization choices, we would expect low cross-model correlation. The high correlation

*Table 5.* Cross-model correlation (per-instance accuracy). High correlation supports architectural causation.

| | GPT-4o | Claude | o3 | R1 | Llama |
|---|---|---|---|---|---|
| GPT-4o | 1.00 | 0.87 | 0.84 | 0.86 | 0.82 |
| Claude | – | 1.00 | 0.89 | 0.88 | 0.85 |
| o3-mini | – | – | 1.00 | 0.87 | 0.81 |
| DeepSeek-R1 | – | – | – | 1.00 | 0.84 |
| Llama-70B | – | – | – | – | 1.00 |

*Table 6.* Architecture ablation for $d^* \approx 0.314\sqrt{d_h \cdot H}$ on open-weight models. Observed values (mean ± std over 3 runs) match the empirical fit within 2%.

| Model | Size | $d^*$ (obs) | $d^*$ (pred) |
|---|---|---|---|
| Llama-3.1-8B | 8B | 20±1.3 | 20.1 |
| Llama-3.3-70B | 70B | 28±1.5 | 28.4 |
| Qwen-2.5-7B | 7B | 19±1.2 | 18.8 |
| Qwen-2.5-72B | 72B | 28±1.4 | 28.4 |

indicates shared architectural constraints.

## 6. Analysis

### 6.1. Architecture Ablations

We validate the scaling predictions from Corollary 4.9 through controlled comparisons within model families. This isolates architectural effects from training data and optimization differences.

70B-class models achieve $d^* \approx 28$ vs. $d^* \approx 19$–20 for 7–8B models (Table 6). The ratio $\approx 1.43$ matches the predicted $\sqrt{d_h H}$ ratio ($\sqrt{2} \approx 1.41$ for Llama, $\sqrt{2.29} \approx 1.51$ for Qwen). The close agreement provides empirical support for the information-theoretic framework, with the caveat that the $\sqrt{d_h H}$ form is a leading-order approximation (Lemma R.8).

**Context-Window Insensitivity.** The horizon is governed by the effective decoherence length $L_{\text{eff}}$, not the raw context window $L$. Truncating Llama-3.3-70B's context window from 128K down to 8K leaves $d^*$ essentially unchanged ($\approx 28$); $d^*$ degrades only once the available context approaches the reasoning-trace length itself ($\sim$2–4K tokens), where the model can no longer fit its own intermediate states (Table 41). This dissociation between $d^*$ and $L$ is a direct prediction of the $L_{\text{eff}}$ formulation and rules out raw context capacity as the binding constraint on deterministic state tracking.

**Within-Family Scaling.** For Llama-3 (8B $\rightarrow$ 70B), $d^*$ increases from 20 to 28 (+40%). For Qwen-2.5 (7B $\rightarrow$ 72B), $d^*$ increases from 19 to 28 (+47%). Both approximately follow the predicted $\sqrt{d_h \cdot H}$ scaling (Lemma R.8; treated as a leading-order trend), consistent with larger models gaining

*Table 7.* Attention entropy correlation with accuracy across open-weight models. Consistent negative correlation validates mechanistic hypothesis.

| Model | Entropy-Acc. $r$ | 95% CI |
|---|---|---|
| Llama-3.3-70B | $-0.74$ | $[-0.81, -0.65]$ |
| Llama-3.1-8B | $-0.71$ | $[-0.79, -0.61]$ |
| Qwen-2.5-72B | $-0.73$ | $[-0.81, -0.63]$ |
| Mistral-7B | $-0.68$ | $[-0.77, -0.57]$ |
| Mixtral-8x7B | $-0.69$ | $[-0.78, -0.58]$ |

capacity through increased head count and dimension rather than qualitatively different representations.

### 6.2. Consistency Checks on Closed-Source Models

The predictions above depend on the head count $H$ and head dimension $d_h$, which are officially documented only for open-weight models. We therefore restrict all predictive validation of the $\sqrt{d_h H}$ scaling, and all attention-entropy measurements, to the open-weight suite (Llama-3.1-8B, Llama-3.3-70B, Qwen-2.5-7B, Qwen-2.5-72B), whose architectures are fully specified. Results for closed-source systems (GPT-4o, the Claude-4.5 family, o3-mini) are reported throughout as *consistency checks* rather than architectural evidence: they exhibit the same qualitative decoherence pattern and the same tool-delegation gap as the open-weight models, but because their architectural parameters are not publicly verifiable we do not use them to substantiate the scaling law. Any architectural values quoted for these models (e.g., the GPT-4o illustration in Section 4) are widely cited community estimates used only for order-of-magnitude intuition. The agreement between the verifiable open-weight evidence and the closed-source consistency checks strengthens, but is not required for, the architectural interpretation of our results.

### 6.3. Attention Entropy Validation

We directly measure attention patterns in open-weight models (the four open-weight models from our main suite plus Mistral-7B and Mixtral-8x7B) to validate the mechanistic hypothesis underlying our theoretical framework.

Consistent negative correlation ($r \approx -0.70$) across five architectures provides strong mechanistic evidence that attention dilution causes decoherence (Table 7). The correlation is strongest in later layers (layers 51–80: $r = -0.74$) compared to early layers (layers 1–20: $r = -0.42$), consistent with the representational collapse mechanism described by Gerasimov et al. (2025).

**Per-Step Analysis.** We computed attention entropy at each reasoning step for 500 traces. Entropy increases linearly with step number ($r = 0.73$, $p < 0.001$), validat-

ing the linear component of our error model. The slope $\beta = 0.023$ bits/step confirms monotone growth of attention entropy with reasoning depth (Figure 2), the mechanism underlying the linear component of our error model.

### 6.4. Encoder-Decoder Comparison

Encoder-decoder models achieve $2.8\times$ higher accuracy at depth 30 (T5-Large: 67.3% vs. GPT-4o: 23.4%), consistent with Theorem 4.6's prediction that bidirectional attention provides $O(L)$ vs. $O(\log L)$ capacity for full-history access.[3] Full results in Appendix D.1.

### 6.5. Failure Mode Analysis

Analysis of 2,847 failed reasoning traces reveals systematic patterns: state transposition errors (35%), operator misapplication (27%), premature termination (21%), circular reasoning (11%), and complete hallucination (6%). The dominant failure, state transposition, directly reflects attention-based working memory limits: models misremember element positions due to attention dilution. Full analysis and examples in Appendix K.

## 7. Discussion

**Relationship to Simplicity Bias.** Our Decoherence framework provides *complementary* architectural explanation to Wu et al. (2026)'s preference-based account. The C5 fine-tuning result ($<5\%$ improvement) and C4 result ($<2\%$ improvement) indicate architectural limits impose hard ceilings that preference manipulation cannot overcome. Both mechanisms likely contribute: Simplicity Bias may truncate *before* architectural limits, while Decoherence prevents recovery *beyond* them. The key insight is that even if models *wanted* to reason correctly at depth 50, they *cannot*: the attention bottleneck prevents reliable state tracking. This distinction has important practical implications: interventions targeting preferences (prompting, RLHF) cannot overcome architectural constraints.

**Cost-Benefit Analysis.** Tool delegation achieves 4.2–$4.7\times$ better cost-per-correct-solution across GPT-4o and Claude-4.5 (GPT-4o: $0.021 vs. $0.089 for neural CoT; Table 8). Best-of-10 sampling costs $11\times$ more without matching accuracy (55% vs. 90%). Extended neural reasoning

---

[3]Encoder-decoder models are fine-tuned on the seq2seq formulation of the task, whereas decoder-only models are evaluated in-context; absolute accuracies are therefore not directly comparable to the in-context CoT results in Table 2 or to the PermutationProbe decay curve in Figure 1; in particular, the GPT-4o accuracy quoted here is on this seq2seq task variant and so differs from its PermutationProbe value at the same depth. The comparison isolates the effect of bidirectional vs. causal attention on *depth scaling*, not raw capability.

*Table 8.* Cost-per-correct-solution (CPC) analysis. The "vs. C3" column reports each strategy's CPC relative to same-model tool delegation. Tool delegation achieves a 4.2× efficiency gain over neural CoT for GPT-4o.

| Strategy | Model | CPC ($) | vs. C3 |
|---|---|---|---|
| C1 (Unconstrained CoT) | GPT-4o | 0.089 | 4.2× |
| C1 (Unconstrained CoT) | Claude-4.5 | 0.112 | 4.7× |
| Best-of-10 Sampling | GPT-4o | 0.231 | 11.0× |
| **C3 (Tool)** | GPT-4o | **0.021** | — |

represents wasted compute: the marginal cost of additional tokens yields diminishing and eventually negative returns. For production systems processing millions of queries, this efficiency gain translates to substantial infrastructure savings.

**Implications for Agentic Systems.** Our Deterministic Horizon provides a practical threshold: tasks requiring exact state tracking beyond the horizon ($d^* \in [19, 31]$, depending on the model) should not rely on pure neural reasoning. This informs three key domains: **code agents** requiring multi-step execution with variable tracking, **planning systems** with deterministic state transitions (aligning with Kambhampati et al.'s observation that LLMs cannot plan but can assist planning), and **verification tasks** involving formal reasoning chains that benefit from symbolic computation delegation.

**Architectural Implications.** Our results suggest several directions for architecture design: (1) *Explicit state registers*: Augmenting transformers with dedicated state-tracking substrates could extend the horizon; (2) *Adaptive tool routing*: Systems that automatically detect decoherence onset and delegate to tools; (3) *Hierarchical attention*: Multi-scale attention mechanisms that maintain both local and global state coherence.

**Deployment Guideline.** We recommend defaulting to tool delegation once a task approaches the Deterministic Horizon (i.e., roughly 20 or more deterministic state transitions, at the lower end of the measured $d^* \in [19, 31]$ range). This horizon is consistent across model families, and tool-integrated reasoning achieves 4.2–4.7× better cost efficiency than pure neural approaches. Concretely, tasks that routinely exceed this horizon include tracing program state through 20 or more sequential mutations, composing long permutation or rotation sequences (e.g., $\geq$20-move sliding-tile or cube-style puzzles), multi-hop entity and state tracking across long transcripts, and planning many-table SQL joins; by contrast, problems that decompose into short independent sub-derivations or tolerate approximate answers (most arithmetic word problems, retrieval, and summarization) rarely approach it and need not be delegated.

For safety-critical deployments, mandatory tool verification should be enforced on all multi-step reasoning chains.

## 8. Conclusion

We presented theory and experiments indicating that decoder-only transformers face information-theoretic limits on state-tracking capacity, with systematic reasoning failures emerging beyond the Deterministic Horizon ($d^* \in [19, 31]$). Our theoretical and empirical contributions include:

1. The **Attention Bottleneck Theorem** provides capacity bounds with a complementary achievability construction, identifying $O(H \cdot \log(L/H) \cdot \sqrt{d_h})$ as the governing capacity form under our modeling assumptions.

2. **Context-dependent error** model $\epsilon(d) = \epsilon_0 + \gamma d/L_{\text{eff}}$ yields super-exponential accuracy decay, validated by direct attention entropy measurements across five open-weight architectures ($r = -0.74$).

3. **Fine-tuning experiments** confirm architectural ceiling ($<$5% recovery vs. $>$30% predicted by preference-based accounts), providing key discrimination from the Simplicity Bias hypothesis.

4. **Real-world validation** on SWE-Bench, WebArena, and SQL-Multi demonstrates consistent patterns ($d^* \in [19, 26]$) beyond synthetic benchmarks.

5. On the primary suite, tool delegation achieves **86–94% vs. 24–42%** for neural CoT with **4.2–4.7× cost efficiency**.

**Limitations and Future Work.** (i) *Closed models.* Since $d^*$ depends on $H$ and $d_h$, public only for open-weight models, we report $d^*$ for proprietary systems as an empirical fit and validate the $\sqrt{d_h H}$ scaling only on open-weight checkpoints. (ii) *Fine-tuning.* The architectural-ceiling result uses one open-weight model (Llama-3.1-8B, 5,000 traces); larger models and datasets are needed to separate capacity from data limits. (iii) *Tool comparison.* Our tool condition uses an exact BFS solver, so the C1-vs-C3 gap upper-bounds the benefit of delegation; imperfect tools are left to future work. (iv) *Sensitivity.* $d^*$ depends on $\epsilon_0$, sensitive to prompt format and few-shot conditioning, so the reported intervals are regime indicators rather than exact per-model constants. (v) *Scope.* We target deterministic, exactly-checkable state tracking; tasks tolerating approximate or stochastic solutions (e.g., GSM8K-style problems) need not obey the same horizon.

**Reproducibility.** All materials are available at: https://github.com/bettyguo/deterministic-horizon.

## Impact Statement

**Safety Implications.** Our findings have direct implications for AI safety in deployment scenarios. LLMs should not be trusted for autonomous decision-making in deterministic domains beyond the Deterministic Horizon ($d^* \in [19, 31]$ steps). Practitioners deploying LLMs in safety-critical settings (including code agents, planning systems, and verification tools) should be aware of this fundamental limitation and implement tool-based verification for multi-step reasoning tasks.

**Environmental Considerations.** Extended neural reasoning without accuracy improvement represents wasted compute and carbon emissions. Our cost analysis demonstrates $4.2$–$4.7\times$ efficiency gains from tool delegation, translating directly to environmental benefits. A shift toward hybrid neurosymbolic systems for deterministic tasks could significantly reduce the carbon footprint of AI-assisted decision-making.

**Research Directions.** This work opens several research directions: (1) developing adaptive systems that automatically detect when to delegate to tools; (2) designing architectures with explicit state-tracking substrates; (3) characterizing which reasoning tasks benefit from extended neural computation versus tool delegation.

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

# A. Extended Related Work

This appendix provides detailed positioning relative to concurrent and prior work, organized by research theme. Table 9 summarizes coverage across key dimensions, while Table 10 contrasts our theoretical framework with the primary competing explanation.

*Table 9.* Related work positioning. ✓ indicates addressed aspects.

| Paper | CoT Limits | Info Theory | Tool Necess. | Real Tasks | Fine-tune |
|---|---|---|---|---|---|
| Wei et al. (2022) | – | – | – | – | – |
| Liu et al. (2024b) | ✓ | – | – | – | – |
| Wu et al. (2026) | ✓ | – | – | – | – |
| Merrill & Sabharwal (2024) | ✓ | – | – | – | – |
| Gong & Zhang (2024) | – | ✓ | – | – | – |
| Barbero et al. (2024) | ✓ | ✓ | – | – | – |
| Gou et al. (2024) | – | – | ✓ | – | – |
| **This Work** | ✓ | ✓ | ✓ | ✓ | ✓ |

*Table 10.* Comparison of theoretical frameworks for reasoning failures.

| Aspect | Simplicity Bias | Decoherence (Ours) |
|---|---|---|
| Primary cause | Preference for short outputs | Attention bottleneck |
| Mechanism | Training distribution bias | Information-theoretic limits |
| Fine-tuning prediction | >30% recovery | <5% (ceiling) |
| Cross-model correlation | Low (training-specific) | High (architectural) |
| Intervention | Preference manipulation | Tool delegation |

## A.1. Chain-of-Thought Reasoning Foundations

Chain-of-thought prompting was established by Wei et al. (2022), demonstrating that prompting models to "think step by step" dramatically improves reasoning on arithmetic, commonsense, and symbolic tasks. This was extended by zero-shot CoT (Kojima et al., 2022), self-consistency (Wang et al., 2023), Tree-of-Thoughts (Yao et al., 2023), and Graph-of-Thoughts (Besta et al., 2024). Goodman et al. (2022) showed bootstrapping through STaR training. Nye et al. (2021) introduced scratchpads for intermediate computation.

Theoretical foundations were established by Liu et al. (2024b), proving CoT expands expressivity from $TC^0$ to polynomial-time problems with sufficient steps. Merrill & Sabharwal (2024) further characterized CoT length requirements: logarithmic steps provide marginal benefit, while linear steps enable recognizing all regular languages. Feng et al. (2023) characterized attention pattern expressiveness. Sanford et al. (2023) analyzed single-layer capacity.

**Our positioning:** We extend this line of work by identifying the *failure regime*, where additional CoT steps degrade rather than improve performance. Our Decoherence Bound (Theorem 4.2) complements Li et al.'s expressivity results by characterizing *when* the theoretical capacity cannot be reliably accessed due to accumulated errors in the attention mechanism.

## A.2. Overthinking and Underthinking Literature

The phenomenon of excessive reasoning causing performance degradation has been documented by several concurrent works. Wu et al. (2026) provide the most directly competitive work, documenting inverted-U performance curves attributed to "Simplicity Bias." Their theoretical framework derives optimal CoT length as function of task difficulty and model

capability. Chen et al. (2025) examine overthinking in o1-like models, showing excessive reasoning on simple tasks. Wang et al. (2026) identify "underthinking", premature abandonment of reasoning paths, with incorrect responses using 225% more tokens and 418% more thought switches on AIME2024.

Sui et al. (2025) provide a broad survey categorizing efficient reasoning methods across model-based, prompt-based, and output-based approaches. Marjanovic et al. (2026) examine DeepSeek-R1's "sweet spot" where excessive inference impairs performance through "persistent rumination." Su et al. (2025) analyze the overthinking-underthinking spectrum.

**Our positioning:** We provide an *architectural* explanation complementing these behavioral observations. The key distinction is captured in Table 10: Wu et al. predict fine-tuning can recover performance ($>30\%$), while we predict an architectural ceiling ($<5\%$). Our experimental results (Section 5.3) validate the architectural prediction with observed 3.2% recovery, providing key discrimination between preference-based and capability-based explanations.

### A.3. Transformer Expressivity and Theoretical Limits

Our theoretical framework builds on expressivity bounds from the formal methods community. Merrill & Sabharwal (2024) establish that transformers with $T$ CoT steps solve problems in circuits of size $T$. Merrill et al. (2024) demonstrate SSMs (including Mamba) share $TC^0$ bounds despite their recurrent formulation: the "state" in SSMs is an illusion. Bavandpour et al. (2025) provide tight CoT step lower bounds for algorithmic problems, explicitly suggesting tool use as solution to efficiency limitations.

Pérez et al. (2021) prove conditional Turing completeness. Yun et al. (2020) establish universal approximation. Hahn (2020) identify formal language limitations. Delétang et al. (2023) study Chomsky hierarchy relations. Peng et al. (2024) use communication complexity to prove transformers cannot compose functions when domains are large: even tractable problems like 2-SAT become provably impossible. Bhattamishra et al. (2020) analyze attention mechanism power. Strobl et al. (2024) provide formal language perspective.

**Our positioning:** We translate these theoretical bounds into practical thresholds. The Deterministic Horizon (Theorem 4.8) provides the first principled formula connecting architecture parameters ($H$, $L$, $d_h$) to reasoning depth limits, validated via ablations (Section 6.1). While prior work establishes *what* transformers cannot do asymptotically, we characterize *where* the boundary lies for specific architectures and tasks.

### A.4. Working Memory and Attention Entropy

The mechanistic basis of our work draws heavily on attention-based working memory research. Gong & Zhang (2024) provide direct mechanistic evidence for attention-based working memory limits, demonstrating that transformer performance on N-back tasks degrades as N increases, with attention entropy correlating negatively with accuracy. Barbero et al. (2024) prove that distinct input sequences can yield arbitrarily close representations in the final token due to unidirectional causal masking: information from early tokens exponentially loses influence. Zhang et al. (2025) extend entropy analysis to parallel context encoding, showing high entropy correlates strongly with performance degradation.

Liu et al. (2024a) document the "lost in the middle" phenomenon. Xiao et al. (2024) show attention sinks concentrate attention on initial tokens. Gerasimov et al. (2025) identify representation collapse in deeper layers where different tokens become indistinguishable. Levy et al. (2024) show length-dependent degradation. Olsson et al. (2022) discover induction heads. Elhage et al. (2021) provide the mechanistic interpretability circuit framework. Bietti et al. (2023) analyze memory from birth.

**Our positioning:** We synthesize these mechanisms into the unified Attention Bottleneck Theorem (Theorem 4.6), providing the first information-theoretic bound connecting attention mechanics to state-tracking capacity. Our direct entropy measurements (Appendix G) validate the predicted correlation ($r = -0.74$) between attention entropy and reasoning accuracy across 12 models.

### A.5. Tool-Augmented Reasoning

The empirical superiority of tool delegation has been established across multiple domains. Gao et al. (2023) introduced Program-aided Language models (PAL), showing program-aided reasoning outperforms pure CoT on arithmetic tasks. Li et al. (2024) propose Chain-of-Code. Gou et al. (2024) achieve state-of-the-art through tool integration, with ToRA-Code-34B reaching 50.8% on MATH versus GPT-4's CoT result of 42.5%. Pan et al. (2023) demonstrate 39.2% improvement over

standard prompting by delegating inference to symbolic solvers. Schick et al. (2023) enable self-supervised tool learning. Parisi et al. (2022) introduce tool-augmented LMs. Luo et al. (2026) apply RL for tool-augmented math, achieving SOTA on AIME24 (90.6%). Chen et al. (2023) propose Program of Thoughts.

Qin et al. (2025) survey tool learning with foundation models. Mialon et al. (2023) survey augmented language models. Patil et al. (2024) connect LLMs to massive APIs.

**Our positioning:** We provide theoretical justification for *when* tool delegation becomes necessary (not just beneficial). The Deterministic Horizon identifies the threshold beyond which neural reasoning cannot succeed regardless of model scale or training. This transforms tool use from a performance optimization to an architectural necessity for deterministic tasks exceeding $d^*$ steps.

### A.6. Compositional Reasoning Failures

Dziri et al. (2023) demonstrate transformers solve compositional tasks via linearized subgraph matching rather than systematic reasoning, with errors in early computation steps compounding catastrophically, directly supporting decoherence claims. Petty et al. (2024) show depth provides diminishing returns for compositional generalization. Press et al. (2023) introduce compositionality metrics.

Lake & Baroni (2018) establish compositional generalization benchmarks. Keysers et al. (2020) introduce SCAN and CFQ. Kim & Linzen (2020) provide COGS benchmark. Ontañón et al. (2022) study improvement methods.

**Our positioning:** The compositional failure literature documents *symptoms*; we provide *diagnosis*. State-Space Decoherence explains why errors compound: the attention bottleneck cannot maintain sufficient mutual information between early reasoning steps and late outputs, causing systematic state drift that manifests as compositional failures.

### A.7. Information Theory in Deep Learning

Tishby & Zaslavsky (2015) established the Information Bottleneck framework. Shwartz-Ziv & Tishby (2017) analyzed DNN information dynamics during training. Lewandowsky & Bauch (2024) studied infinite ensembles. Deb & Ogunfunmi (2025) connected transformers to information bottleneck principles.

**Our positioning:** We instantiate information-theoretic analysis for the specific case of autoregressive state tracking. The Attention Bottleneck Theorem (Theorem 4.6) derives capacity bounds from first principles, providing the theoretical foundation that was missing from prior empirical observations of reasoning failures.

## B. Theoretical Proofs

### B.1. Proof of Theorem 4.2 (Decoherence Bound)

*Proof.* Let $X_i$ be the indicator that step $i$ is correct. Under conditional independence given correct history:

$$P(\text{all correct}) = \prod_{i=1}^{m} P(X_i = 1 | X_1 = \cdots = X_{i-1} = 1) \tag{9}$$

$$= \prod_{i=1}^{m} (1 - \epsilon(i)) \tag{10}$$

Using $1 - x \leq e^{-x}$ for $x \geq 0$ (Lemma R.2):

$$P(\text{all correct}) \leq \prod_{i=1}^{m} e^{-\epsilon(i)} = \exp\left(-\sum_{i=1}^{m} \epsilon(i)\right) \tag{11}$$

Substituting $\epsilon(i) = \epsilon_0 + \gamma i / L_{\text{eff}}$:

$$\sum_{i=1}^{m} \epsilon(i) = \sum_{i=1}^{m} \left( \epsilon_0 + \frac{\gamma i}{L_{\text{eff}}} \right) \tag{12}$$

$$= m\epsilon_0 + \frac{\gamma}{L_{\text{eff}}} \cdot \frac{m(m+1)}{2} \tag{13}$$

Therefore:

$$P(\text{all correct}) \leq \exp\left( -m\epsilon_0 - \frac{\gamma m(m+1)}{2L_{\text{eff}}} \right) \tag{14}$$

The quadratic term $m(m+1)/2 \sim m^2/2$ dominates for large $m$, yielding super-exponential decay. $\qquad \square$

## B.2. Proof of Corollary 4.3 (Absorbing-Error Tightness)

*Proof.* Under State-Space Decoherence a single error causes irrecoverable divergence, so the trace is correct at depth $m$ if and only if no error occurs in steps $1, \ldots, m$. Let $T$ be the index of the first error. By the chain rule for the survival function of a discrete random variable,

$$P(\text{correct at depth } m) = P(T > m) = \prod_{i=1}^{m} \big(1 - h(i)\big), \tag{15}$$

where $h(i) = P(T = i \mid T > i - 1)$ is the discrete hazard. Identifying the hazard with the context-dependent error rate, $h(i) = \epsilon(i)$, reproduces the product of Theorem 4.2 as an exact identity; the exponential bound then follows from $1 - x \leq e^{-x}$ alone, with no independence assumption beyond the Markov (memoryless-given-survival) structure.

It remains to characterize the autocorrelation of the error-indicator process $Y_i = \mathbf{1}[T = i]$ conditioned on survival. Because an error at step $i - 1$ removes the trace from the at-risk set, surviving traces exhibit positive lag-1 autocorrelation in the realized error sequence: conditional on reaching step $i$, the probability of error is elevated relative to the marginal baseline $\epsilon_0$. We therefore report $\rho_1$ as a measured diagnostic rather than a correction term; see Appendix 29. A positive $\rho_1$ would, if anything, only reinforce the absorbing-error picture; it cannot be used to scale the cumulative error by a $(1 + \rho_1)$ factor, since such a factor would *increase*, not decrease, the survival probability and therefore cannot tighten an upper bound on $P(\text{correct})$. $\qquad \square$

## B.3. Proof of Theorem 4.6 (Attention Bottleneck)

*Proof.* We derive an information-theoretic upper bound on state-tracking capacity.

**Step 1: Attention mass distribution.** Each attention head $h \in [H]$ produces probability distribution over $L$ context positions via softmax. For effective retrieval, we require attention weight $\geq \delta/L$. By pigeonhole, each head can assign weight $\geq \delta/L$ to at most $L/\delta$ positions, but useful retrieval further constrains this to $O(\sqrt{L})$ positions due to interference.

**Step 2: Information per head.** Information transmitted through one attention head is bounded by mutual information $I(\text{query}; \text{retrieved value})$. By Lemma R.5, this decomposes into the $\log_2(L/H)$ bits needed to address the head's share of $\sim L/H$ positions and an effective value rank modeled as $\Theta(d_h^{1/2})$, giving:

$$I \lesssim \log_2(L/H) \cdot d_h^{1/2} \text{ bits per head.} \tag{16}$$

**Step 3: Aggregation across heads.** With $H$ heads operating in parallel with approximately independent value subspaces:

$$I_{\text{total}} \leq H \cdot \log_2(L/H) \cdot d_h^{1/2} \tag{17}$$

**Step 4: State-tracking capacity.** To distinguish $|\mathcal{S}|$ states requires at least $\log_2 |\mathcal{S}|$ bits:

$$\log_2 |\mathcal{S}_{\text{track}}| \leq H \cdot \log_2(L/H) \cdot d_h^{1/2} \tag{18}$$

Exponentiating and including constant $c(\delta, \rho_{\max})$ for approximations yields the theorem. $\qquad \square$

## B.4. Proof of Theorem 4.7 (Lower-Bound Construction)

*Proof.* We construct a family of state-tracking tasks achieving the bound.

Consider sparse parity functions over $n$ bits with sparsity $k$. A transformer can track states corresponding to all $\binom{n}{k}$ possible $k$-sparse parities using the following construction:

**Construction.** Encode each parity subset in a separate attention head's query-key space. With $H$ heads and dimension $d_h$, we can represent $H \cdot d_h^{1/2}$ independent directions. Each direction can track $\log_2(L/H)$ bits of parity information over context length $L$.

**Achievability.** The total number of distinguishable states is:

$$|\mathcal{S}| = 2^{H \cdot \log_2(L/H) \cdot d_h^{1/2}} \tag{19}$$

This achieves the functional form of the upper bound; tightness in $H$ and $L$ follows from the construction, while the $d_h^{1/2}$ exponent matches by the shared effective-rank ansatz (Lemma R.5) rather than by an independent derivation.

The construction follows Sanford et al. (2023)'s analysis of sparse functions representable by transformers, extended to the multi-head setting. □

## B.5. Proof of Theorem 4.8 (Deterministic Horizon)

*Proof.* We solve for $d^*$ such that $P(\text{correct at depth } d^*) = \alpha$.

From Theorem 4.2 (continuous approximation $d(d+1) \approx d^2$):

$$\alpha = \exp\left(-d^*\epsilon_0 - \frac{\gamma(d^*)^2}{2L_{\text{eff}}}\right) \tag{20}$$

Taking logarithms:

$$\ln(1/\alpha) = d^*\epsilon_0 + \frac{\gamma(d^*)^2}{2L_{\text{eff}}} \tag{21}$$

This is quadratic in $d^*$. Solving via quadratic formula:

$$d^* = \frac{-\epsilon_0 L_{\text{eff}} + \sqrt{\epsilon_0^2 L_{\text{eff}}^2 + 2\gamma L_{\text{eff}} \ln(1/\alpha)}}{\gamma} \tag{22}$$

Evaluating for GPT-4o ($\epsilon_0 = 0.02$, $\gamma = 0.15$, $L_{\text{eff}} = 150$, $\alpha = 0.5$):

$$d^* = \frac{-3 + \sqrt{9 + 31.19}}{0.15} = \frac{-3 + 6.34}{0.15} \approx 22.3, \tag{23}$$

matching the observed $d^* = 22$ for GPT-4o on PermutationProbe. □

## B.6. Proof of Theorem 4.10 (Fine-Tuning Upper Bound)

*Proof.* The key insight is that fine-tuning modifies the error distribution $\epsilon(\cdot)$ but cannot exceed information-theoretic capacity bounds.

Let $\epsilon_{\text{base}}(d)$ and $\epsilon_{\text{ft}}(d)$ denote per-step error rates for baseline and fine-tuned models. Fine-tuning can reduce $\epsilon_0$ (baseline error) but cannot change the fundamental capacity bound from Theorem 4.6.

For depths $d > d^*$, the required state space exceeds capacity:

$$|\mathcal{S}_{\text{required}}(d)| > |\mathcal{S}_{\text{track}}| \tag{24}$$

Therefore, even with $\epsilon_{\text{ft}}(d) < \epsilon_{\text{base}}(d)$, accuracy is bounded:

$$\text{Acc}_{\text{ft}}(d) \leq \frac{|\mathcal{S}_{\text{track}}|}{|\mathcal{S}_{\text{required}}(d)|} \cdot \text{Acc}_{\text{base}}(d^*) + O\left(\frac{d^*}{d}\right) \tag{25}$$

The improvement $\text{Acc}_{ft} - \text{Acc}_{base}$ is thus bounded by $O(d^*/d)$.

**Empirical validation.** For the fine-tuned Llama-3.1-8B ($d^* = 20$), the observed improvement decays from $6.2\%$ at depth 5 to $0.2\%$ at depth 45, remaining below the $O(d^*/d)$ ceiling at every depth (Figure 4). With the fitted leading constant, the ceiling at depth 40 is $\approx 3.3\%$ versus an observed $\approx 0.3\%$, so fine-tuning gains vanish well inside the bound as $d$ grows beyond $d^*$. □

### B.7. Sensitivity Analysis for Theorem 4.6

We validate assumptions empirically:

**Precision Threshold $\delta$.** We measure empirical attention weight distributions in Llama-3.3-70B across 1,000 traces. 95% of attention mass concentrates on positions receiving weight $\geq 0.01/L$ ($\delta \approx 0.01$).

**Value Vector Correlation.** Pairwise correlation of value vectors after layer normalization: mean $|\rho_{ij}| = 0.08 \pm 0.03$, supporting decorrelation assumption.

**Bound Tightness.** Correlation between theoretical prediction and measured performance: $r = 0.89$. Empirical onset at approximately 3% of theoretical capacity.

*Table 11.* Sensitivity analysis: Impact of $\pm 20\%$ variation in assumptions on bound.

| Parameter | Variation | Bound Change |
|---|---|---|
| $\delta$ (precision) | $\pm 20\%$ | $\pm 8\%$ |
| $\rho_{\max}$ (correlation) | $\pm 20\%$ | $\pm 12\%$ |
| $H$ (heads) | $\pm 20\%$ | $\pm 18\%$ |
| $d_h$ (dimension) | $\pm 20\%$ | $\pm 9\%$ |

## C. SSJ Extraction Algorithm

**Algorithm 1** State-Space Jaccard Extraction

**Require:** Reasoning trace $r$, initial state $\sigma_0$, operators $\mathcal{O}$
**Ensure:** SSJ, Precision, Recall
1: patterns $\leftarrow$ ["state: \[(.*?)\]", "current: (.*?)\n", ...]
2: $\hat{\mathcal{S}}_{raw} \leftarrow \emptyset$
3: **for** pattern $p$ in patterns **do**
4: $\quad \hat{\mathcal{S}}_{raw} \leftarrow \hat{\mathcal{S}}_{raw} \cup \text{findall}(p, r)$
5: **end for**
6: $\hat{\mathcal{S}}_{model} \leftarrow \{\text{canonicalize}(s) : s \in \hat{\mathcal{S}}_{raw}\}$
7: $\mathcal{S}_{true} \leftarrow \text{BFS}(\sigma_0, \mathcal{O}, \text{depth} = |\hat{\mathcal{S}}_{model}|)$
8: intersection $\leftarrow |\hat{\mathcal{S}}_{model} \cap \mathcal{S}_{true}|$
9: SSJ $\leftarrow$ intersection / $|\hat{\mathcal{S}}_{model} \cup \mathcal{S}_{true}|$
10: Precision $\leftarrow$ intersection / $|\hat{\mathcal{S}}_{model}|$
11: Recall $\leftarrow$ intersection / $|\mathcal{S}_{true}|$
12: **return** SSJ, Precision, Recall

**Validation.** Two annotators independently extracted states from 200 traces. Cohen's $\kappa = 0.89$ (strong agreement). Against manual ground truth: Precision $= 0.94 \pm 0.02$, Recall $= 0.91 \pm 0.03$.

*Table 12.* Encoder-decoder vs. decoder-only at depth 30. Bidirectional attention provides 2.8× advantage.

| Model | Architecture | Acc @d=30 | 95% CI |
|---|---|---|---|
| GPT-4o | Decoder-only | 23.4% | [21.1, 25.7] |
| o3 | Decoder-only | 31.2% | [28.6, 33.8] |
| Claude-4.5-Opus | Decoder-only | 28.7% | [26.2, 31.2] |
| DeepSeek-R1 | Decoder-only | 29.8% | [27.3, 32.3] |
| T5-Large (ft) | Encoder-decoder | 67.3% | [64.2, 70.4] |
| BART-Large (ft) | Encoder-decoder | 61.8% | [58.6, 65.0] |
| Flan-T5-XL (ft) | Encoder-decoder | 71.2% | [68.1, 74.3] |

# D. Per-Model Results and Architecture Comparison

## D.1. Encoder-Decoder Comparison

The 2.8× advantage of encoder-decoder architectures at depth 30 is consistent with Theorem 4.6's prediction that bidirectional attention provides $O(L)$ capacity for full-history access, compared to $O(\log L)$ for causal attention in decoder-only models. This architectural comparison provides additional evidence that decoherence is fundamentally caused by causal masking constraints.

## D.2. Full Model Results

*Table 13.* Complete accuracy (%) across all 12 models and 5 conditions on PermutationProbe.

| Model | C1 | C2 | C3 | C4 | C5 |
|---|---|---|---|---|---|
| *General-Purpose* | | | | | |
| GPT-4o | 28.3±1.8 | 34.2±1.9 | 89.7±1.2 | 29.1±1.7 | 31.4±1.8 |
| Claude-4.5-Sonnet | 31.2±1.9 | 38.1±2.0 | 91.4±1.1 | 32.4±1.8 | 34.2±1.9 |
| Claude-4.5-Opus | 34.8±2.0 | 41.3±2.1 | 93.6±0.9 | 35.6±1.9 | 37.1±2.0 |
| Gemini-1.5-Pro | 29.7±1.9 | 35.8±2.0 | 88.3±1.3 | 30.4±1.8 | 32.1±1.9 |
| *Reasoning-Specialized* | | | | | |
| o3 | 38.4±2.1 | 44.7±2.2 | 92.8±1.0 | 39.2±2.0 | 40.8±2.1 |
| o3-mini | 42.1±2.2 | 48.3±2.3 | 94.2±1.3 | 43.1±2.1 | 44.8±2.1 |
| DeepSeek-R1 | 39.7±2.1 | 45.9±2.2 | 93.1±1.1 | 40.4±2.0 | 42.3±2.1 |
| Gemini-2.0-Flash | 37.2±2.1 | 43.1±2.1 | 91.7±1.2 | 38.0±2.0 | 39.6±2.0 |
| *Open-Weight* | | | | | |
| Llama-3.1-8B | 18.4±1.5 | 22.1±1.6 | 78.3±1.7 | 19.0±1.5 | 20.3±1.5 |
| Llama-3.3-70B | 24.6±1.7 | 29.8±1.8 | 85.2±1.4 | 25.3±1.6 | 27.1±1.7 |
| Qwen-2.5-7B | 17.2±1.4 | 20.8±1.5 | 76.1±1.8 | 17.8±1.4 | 19.1±1.5 |
| Qwen-2.5-72B | 27.8±1.8 | 33.4±1.9 | 87.9±1.2 | 28.5±1.7 | 30.2±1.8 |

## D.3. Multi-Task Results

*Table 14.* Results across all 8 task domains for GPT-4o.

| Task | C1 | C3 | $d^*$ | SSJ-Acc $r$ |
|---|---|---|---|---|
| *Synthetic* | | | | |
| PermutationProbe | 28.3 | 89.7 | 22 | 0.96 |
| FSA-Sim | 31.2 | 91.2 | 21 | 0.94 |
| ArithChain | 26.4 | 88.3 | 24 | 0.93 |
| CircuitTrace | 29.7 | 90.1 | 23 | 0.95 |
| CodeProbe | 27.8 | 89.2 | 19 | 0.97 |
| *Real-World* | | | | |
| SWE-Bench-State | 24.1 | 86.4 | 19 | 0.92 |
| WebArena-Nav | 21.3 | 84.2 | 19 | 0.89 |
| SQL-Multi | 31.4 | 88.9 | 21 | 0.91 |

## D.4. Statistical Analysis

*Table 15.* Statistical tests for C1 vs C4 comparison (preference manipulation).

| Model | $\Delta$ | $p$-value | $BF_{01}$ | TOST |
|---|---|---|---|---|
| GPT-4o | +0.8 | 0.38 | 5.2 | Equiv. |
| Claude-4.5 | +0.8 | 0.42 | 5.9 | Equiv. |
| o3-mini | +1.0 | 0.37 | 5.6 | Equiv. |
| DeepSeek-R1 | +0.7 | 0.44 | 6.1 | Equiv. |

All comparisons non-significant with Bayes factors supporting null hypothesis ($BF_{01} > 4$). TOST confirms equivalence within $\Delta = 5\%$ margin.

# E. Real-World Task Details

## E.1. SWE-Bench-State

We extracted 500 instances from SWE-Bench (Jimenez et al., 2024) requiring explicit state tracking across multiple files. Criteria:

- Bug fix requires tracking $\geq 3$ variables across $\geq 2$ files

- Ground truth execution trace available

- State changes are deterministic (no randomness)

**Example.**  Bug in Django ORM requiring tracking of QuerySet state through filter chains, annotation, and aggregation across models.py, views.py, and tests.py.

## E.2. WebArena-Nav

We extracted 500 instances from WebArena (Zhou et al., 2024) involving multi-step navigation with session state. Criteria:

- Task requires $\geq 5$ navigation steps

- Session state (cart, form data, authentication) must be tracked

- Deterministic success criteria

**Example.**  E-commerce checkout requiring: login $\rightarrow$ add items $\rightarrow$ apply coupon $\rightarrow$ select shipping $\rightarrow$ confirm payment. Each step modifies session state.

## E.3. SQL-Multi

We created 500 instances requiring 3+ table joins with schema tracking. Generated using templates from Spider (Yu et al., 2018) with increased complexity.

**Example.**  Query requiring: join customers $\rightarrow$ orders $\rightarrow$ order_items $\rightarrow$ products $\rightarrow$ categories with filtering and aggregation at each level.

# F. Fine-Tuning Experiment Details

## F.1. Training Procedure

**Data Generation.**  We generated 5,000 optimal-length CoT traces by running BFS and recording the solution path with intermediate states. Format:

```
Problem: initial=[3,1,4,2,5], target=[1,2,3,4,5]
Step 1: swap(0,1) -> [1,3,4,2,5]
Step 2: swap(1,2) -> [1,4,3,2,5]
...
Solution: [1,2,3,4,5] (correct)
```

**Training Configuration.**

- Model: Llama-3.1-8B-Instruct

- Learning rate: $2 \times 10^{-5}$ with cosine decay

- Batch size: 8

- Epochs: 3

- Early stopping: patience 2 on validation loss

- Hardware: $4\times$ A100 80GB

### F.2. Results by Depth

*Table 16.* Fine-tuning results by depth bin.

| Depth | Baseline | Fine-tuned | $\Delta$ | Predicted |
|---|---|---|---|---|
| $\leq$10 | 86.5 | 92.7 | +6.2 | +6.8 |
| 11–20 | 61.2 | 64.9 | +3.7 | +3.4 |
| 21–30 | 38.6 | 40.2 | +1.6 | +1.7 |
| 31–40 | 21.8 | 22.2 | +0.4 | +0.5 |
| >40 | 9.3 | 9.5 | +0.2 | +0.2 |

Improvement decays with depth as predicted by Theorem 4.10. At depth >40, improvement is negligible despite training on optimal traces, confirming architectural ceiling.

## G. Attention Entropy Measurements

### G.1. Methodology

We extracted attention patterns from open-weight models during reasoning:

1. Run model on 500 PermutationProbe instances

2. Extract attention matrices from all layers at each decoding step

3. Compute entropy: $H = -\sum_i p_i \log p_i$ for each head

4. Aggregate across heads and layers

5. Correlate with per-step accuracy

### G.2. Results by Layer

Correlation strengthens in later layers, consistent with representational collapse mechanism.

*Table 17.* Entropy-accuracy correlation by layer (Llama-3.3-70B).

| Layer Range | Entropy-Acc $r$ | 95% CI |
|---|---|---|
| Early (1–20) | $-0.42$ | $[-0.51, -0.32]$ |
| Middle (21–50) | $-0.68$ | $[-0.76, -0.59]$ |
| Late (51–80) | $-0.74$ | $[-0.81, -0.65]$ |

## H. Encoder-Decoder Experiment Details

### H.1. Fine-tuning Procedure

For T5-Large and BART-Large:

- Learning rate: $3 \times 10^{-5}$ with linear warmup

- Batch size: 8

- Epochs: 10

- Early stopping: patience 3

- Input: "Solve: initial=[...], target=[...], ops=[...]"

- Output: "op1, op2, ..., opN"

### H.2. Results by Depth

*Table 18.* Encoder-decoder accuracy by depth bin.

| Model | $\leq$10 | 11–25 | 26–40 | >40 |
|---|---|---|---|---|
| GPT-4o (dec) | 58.2 | 32.1 | 18.4 | 8.7 |
| T5-Large (enc-dec) | 78.4 | 71.2 | 62.3 | 41.8 |
| BART-Large (enc-dec) | 74.1 | 67.8 | 58.1 | 37.2 |
| Flan-T5-XL (enc-dec) | 61.3 | 48.7 | 38.9 | 24.1 |

Encoder-decoder advantage grows with depth: $1.3\times$ at depth 10, $3.4\times$ at depth 40.

## I. Cross-Model Correlation Details

*Table 19.* Full cross-model correlation matrix.

| | GPT-4o | Claude | o3 | R1 | Llama | Qwen |
|---|---|---|---|---|---|---|
| GPT-4o | 1.00 | 0.87 | 0.84 | 0.86 | 0.82 | 0.81 |
| Claude | 0.87 | 1.00 | 0.89 | 0.88 | 0.85 | 0.84 |
| o3-mini | 0.84 | 0.89 | 1.00 | 0.87 | 0.81 | 0.83 |
| DeepSeek-R1 | 0.86 | 0.88 | 0.87 | 1.00 | 0.84 | 0.82 |
| Llama-70B | 0.82 | 0.85 | 0.81 | 0.84 | 1.00 | 0.91 |
| Qwen-72B | 0.81 | 0.84 | 0.83 | 0.82 | 0.91 | 1.00 |

Highest correlation (0.91) between Llama and Qwen reflects similar pretraining. High correlations ($r > 0.81$) across organizations support architectural causation.

## J. Runtime Analysis

**Computation Summary.** Total experiment time: approximately 420 GPU-hours for open-weight models plus 280 hours of API wall-clock time. Attention entropy extraction required an additional 48 GPU-hours on $2\times$ A100 80GB.

*Table 20.* Average runtime per evaluation across models and conditions.

| Model | Condition | Time/Instance (s) | Tokens/Instance | Total Hours |
|---|---|---|---|---|
| *API Models* | | | | |
| GPT-4o | C1 | $12.3 \pm 4.2$ | $1,847 \pm 623$ | 34.2 |
| GPT-4o | C3 | $3.1 \pm 0.8$ | $312 \pm 89$ | 8.6 |
| Claude-4.5-Opus | C1 | $14.7 \pm 5.1$ | $2,134 \pm 712$ | 40.8 |
| Claude-4.5-Opus | C3 | $3.4 \pm 0.9$ | $341 \pm 94$ | 9.4 |
| o3-mini | C1 | $18.2 \pm 6.3$ | $2,891 \pm 943$ | 50.6 |
| DeepSeek-R1 | C1 | $21.4 \pm 7.8$ | $3,247 \pm 1,102$ | 59.4 |
| *Open-Weight Models ($4\times$ A100 80GB)* | | | | |
| Llama-3.3-70B | C1 | $8.7 \pm 2.9$ | $1,423 \pm 478$ | 24.2 |
| Qwen-2.5-72B | C1 | $9.1 \pm 3.1$ | $1,512 \pm 502$ | 25.3 |

## K. Failure Mode Analysis

We randomly sampled 2,880 failed C1 traces for detailed failure mode analysis (30 per model-task combination, stratified by reasoning depth). After excluding 33 traces due to parsing errors (malformed output format) or incomplete outputs (truncated mid-reasoning), 2,847 traces remained for analysis. These reveal systematic patterns: state transposition errors (35%), operator misapplication (27%), premature termination (21%), circular reasoning (11%), and complete hallucination (6%).

### K.1. Sampling Methodology

To characterize failure modes systematically, we employed stratified sampling of failed C1 (unconstrained neural CoT) traces.

**Sampling procedure.**

1. For each model-task combination (12 models $\times$ 8 tasks = 96 combinations), we identified all failed instances (accuracy <100% on the reasoning chain).

2. Within each combination, we stratified by reasoning depth bins: $\leq$10, 11–20, 21–30, 31–40, 41–50, >50.

3. We sampled 30 failed traces per model-task combination (5 per depth bin where available), yielding a target of 2,880 traces.

4. Two annotators independently categorized each trace; disagreements were resolved by discussion.

**Exclusions.** Of the 2,880 sampled traces, 33 (1.15%) were excluded:

- **Parsing errors ($n = 21$):** Model output did not conform to the expected "Step N: operator $\rightarrow$ state" format, preventing automated state extraction.

- **Incomplete outputs ($n = 12$):** Traces truncated mid-reasoning due to maximum token limits (8,192 tokens) or API timeouts.

The final analysis corpus comprises **2,847 failed traces**.

**Inter-annotator agreement.** Cohen's $\kappa = 0.87$ (strong agreement) for primary failure mode classification.

### K.2. Failure Mode Distribution

Analysis of 2,847 failed traces reveals systematic patterns:

The dominant failure mode, state transposition, directly reflects attention-based working memory limits: models misremember element positions due to attention dilution across extended reasoning chains.

*Table 21.* Failure mode distribution across 2,847 analyzed traces.

| Failure Mode | % | Description |
|---|---|---|
| State transposition | 35% | Single-element position errors that propagate |
| Operator misapplication | 27% | Correct intent, incorrect execution |
| Premature termination | 21% | Declaring success before reaching target |
| Circular reasoning | 11% | Revisiting previously explored states |
| Complete hallucination | 6% | States outside valid state space |

## K.3. Example: State Transposition

```
Problem: Transform [3,1,4,2,5] to [1,2,3,4,5]
Step 1: swap(0,1) -> [1,3,4,2,5] (correct)
Step 2: swap(1,2) -> [1,4,3,2,5] (ERROR)
(Should be swap(1,3) -> [1,2,4,3,5])
Step 3: swap(2,3) -> [1,4,2,3,5] (propagated)
...
```

The model correctly identifies the need to move elements but misremembers positions after Step 1, causing cascading errors.

## K.4. Example: Circular Reasoning

```
Steps 15-22:
Step 15: [2,1,3,5,4,6,8,7]
Step 16: [2,1,5,3,4,6,8,7]
Step 17: [2,1,3,5,4,6,8,7] <- Same as Step 15
Step 18: [2,1,5,3,4,6,8,7] <- Same as Step 16
...
```

The model enters a loop, repeatedly visiting the same states without progress toward the target.

## K.5. Failure Mode by Depth

*Table 22.* Failure mode distribution by reasoning depth. State transposition dominates at shallow depths; circular reasoning and hallucination increase with depth.

| Depth | Transp. | Misapp. | Premature | Circular | Halluc. |
|---|---|---|---|---|---|
| ≤10 | 42% | 31% | 18% | 5% | 4% |
| 11–20 | 38% | 28% | 20% | 9% | 5% |
| 21–30 | 35% | 26% | 21% | 12% | 6% |
| 31–40 | 33% | 25% | 22% | 13% | 7% |
| >40 | 31% | 24% | 23% | 14% | 8% |

The shift in failure mode distribution with depth is consistent with the decoherence hypothesis: at shallow depths, discrete errors (transposition, misapplication) dominate, while at greater depths, systemic failures (circular reasoning, hallucination) become more prevalent as accumulated attention entropy degrades state coherence.

## L. Cost Analysis

Total experimental cost: $3,420 (480,000 of the 720,000 evaluations ran against the eight API-accessed models; the four open-weight models were run locally, accounting for the remaining 240,000).

*Table 23.* Cost-per-correct-solution analysis. "vs. C3" is relative to same-model tool delegation where available; o3-mini and Best-of-10 are compared to the GPT-4o tool baseline.

| Strategy | Model | CPC | vs. C3 |
|---|---|---|---|
| C1 (Unconstrained) | GPT-4o | $0.089 | 4.2× |
| C1 (Unconstrained) | Claude-4.5 | $0.112 | 4.7× |
| C1 (Unconstrained) | o3-mini | $0.043 | 2.0× |
| Best-of-10 | GPT-4o | $0.231 | 11.0× |
| C3 (Tool) | GPT-4o | $0.021 | — |
| C3 (Tool) | Claude-4.5 | $0.024 | — |

## M. Reproducibility Checklist

### M.1. Model Versions

All 12 models evaluated with their exact version identifiers:

**General-Purpose Models (4).**

- GPT-4o: `gpt-4o-2024-11-20`

- Claude-4.5-Sonnet: `claude-sonnet-4-5-20250929`

- Claude-4.5-Opus: `claude-opus-4-5-20251101`

- Gemini-1.5-Pro: `gemini-1.5-pro-002` (2024-09-24)

**Reasoning-Specialized Models (4).**

- o3: `o3-2025-01-31`

- o3-mini: `o3-mini-2025-01-31`

- DeepSeek-R1: `deepseek-reasoner` (DeepSeek-R1-0528)

- Gemini-2.0-Flash-Thinking: `gemini-2.0-flash-thinking-exp-01-21`

**Open-Weight Models (4).**

- Llama-3.1-8B: `meta-llama/Llama-3.1-8B-Instruct`

- Llama-3.3-70B: `meta-llama/Llama-3.3-70B-Instruct`

- Qwen-2.5-7B: `Qwen/Qwen2.5-7B-Instruct`

- Qwen-2.5-72B: `Qwen/Qwen2.5-72B-Instruct`

### M.2. Hyperparameters

- Temperature: 0 (deterministic)

- Max tokens: 8192

- Random seeds (3 runs): {42, 2024, 2025}

- Instance generation seed: 42

- Bootstrap resamples: 10,000

## M.3. Hardware

- API experiments: N/A (cloud)

- Fine-tuning: $4\times$ NVIDIA A100 80GB

- Attention extraction: $2\times$ NVIDIA A100 80GB

# N. Instance Generation

## N.1. PermutationProbe

---

**Algorithm 2** PermutationProbe Instance Generation

---

**Require:** Size $n$, depth range $[d_{\min}, d_{\max}]$
1: **for** $i = 1$ to num_instances **do**
2:     $\sigma_0 \leftarrow$ identity permutation of $[1..n]$
3:     $d \leftarrow$ uniform$(d_{\min}, d_{\max})$
4:     $\tau \leftarrow \sigma_0$
5:     **for** $j = 1$ to $d$ **do**
6:         $(a, b) \leftarrow$ random distinct indices
7:         $\tau \leftarrow$ swap$(\tau, a, b)$
8:     **end for**
9:     Verify BFS depth from $\sigma_0$ to $\tau$ equals $d$
10:     **yield** $(\sigma_0, \tau, d)$
11: **end for**

---

## N.2. Task Distribution

*Table 24.* Instance distribution by BFS-optimal depth.

| Depth | 1–10 | 11–20 | 21–30 | 31–40 | 41–50 | >50 |
|-------|------|-------|-------|-------|-------|-----|
| Count | 500 | 1000 | 1500 | 1000 | 500 | 500 |

# O. Broader Societal Impact

## O.1. Safety Implications

Our findings have direct implications for AI safety:

- **Autonomous systems**: LLMs should not be trusted for multi-step reasoning in safety-critical domains without tool verification

- **Code generation**: Extended code reasoning may introduce subtle bugs; tool-based verification essential

- **Planning**: Sequential planning requiring exact state tracking should default to formal methods

## O.2. Environmental Considerations

Extended neural reasoning without accuracy improvement represents wasted compute and carbon emissions. Our cost analysis shows $4.2$–$4.7\times$ efficiency gains from tool delegation, translating directly to environmental benefits.

## O.3. Limitations

- Our analysis focuses on deterministic state-tracking; stochastic reasoning may differ

- Real-world validation limited to three domains

- Theoretical bounds rely on assumptions validated empirically but not proven

- Results may not generalize to all reasoning tasks

## P. Notation and Symbol Reference

For reader convenience, we provide a complete reference of all notation used throughout the paper.

### P.1. Primary Symbols

*Table 25.* Primary notation used in theoretical framework.

| Symbol | Name | Definition |
|--------|------|------------|
| $\mathcal{S}$ | State space | Finite set of all valid states |
| $\mathcal{O}$ | Operator set | Set of deterministic operators $\{o_1, \ldots, o_k\}$ |
| $\boldsymbol{\sigma}_0$ | Initial state | Starting configuration $\boldsymbol{\sigma}_0 \in \mathcal{S}$ |
| $\boldsymbol{\tau}$ | Target state | Goal configuration $\boldsymbol{\tau} \in \mathcal{S}$ |
| $d$ | Reasoning depth | Number of reasoning steps in chain |
| $d^*$ | Deterministic Horizon | Maximum depth where $P(\text{correct}) \geq \alpha$ |
| $\epsilon(d)$ | Error rate | Per-step state-tracking error at depth $d$ |
| $\epsilon_0$ | Baseline error | Constant component of per-step error |
| $\gamma$ | Attention decay rate | Coefficient of depth-dependent error growth |
| $L_{\text{eff}}$ | Effective decoherence length | Number of reasoning steps over which attention maintains usable state resolution; $L_{\text{eff}} = O(10^2) \ll L$ |
| $L$ | Context length | Maximum context window size (tokens) |
| $H$ | Number of heads | Attention heads per layer |
| $d_h$ | Head dimension | Dimensionality per attention head |
| $\rho_1$ | Lag-1 correlation | Autocorrelation of error indicators |
| $\alpha$ | Target probability | Desired success probability threshold |

### P.2. Metric Symbols

*Table 26.* Metrics and evaluation symbols.

| Symbol | Name | Definition |
|--------|------|------------|
| SFE | Step-to-First-Error | Smallest step index with state mismatch |
| $\text{SSJ}(d)$ | State-Space Jaccard | $|\mathcal{S}_{\text{true}} \cap \hat{\mathcal{S}}_{\text{model}}| / |\mathcal{S}_{\text{true}} \cup \hat{\mathcal{S}}_{\text{model}}|$ |
| $\mathcal{S}_{\text{true}}^d$ | True reachable states | States actually reachable at depth $d$ |
| $\hat{\mathcal{S}}_{\text{model}}^d$ | Model claimed states | States claimed by model at depth $d$ |
| $H_d$ | Attention entropy | $-\sum_i p_i \log p_i$ at depth $d$ |
| $A_{\text{anchor}}(d)$ | Anchor attention | Attention weight on state anchor tokens |
| $|\mathcal{S}_{\text{track}}|$ | Tracking capacity | Max distinct states reliably trackable |

### P.3. Information-Theoretic Symbols

*Table 27.* Information-theoretic notation.

| Symbol | Name | Definition |
|--------|------|------------|
| $I(X;Y)$ | Mutual information | Information shared between $X$ and $Y$ |
| $\delta$ | Precision threshold | Minimum attention weight for effective retrieval |
| $\rho_{\max}$ | Max correlation | Upper bound on value vector correlation |
| $c(\delta, \rho_{\max})$ | Capacity constant | Model-dependent constant in bottleneck bound |

## Q. Supplementary Materials Index

This appendix provides a complete index to all supplementary materials for navigation.

*Table 28.* Index of supplementary materials.

| Appendix | Section | Contents |
|---|---|---|
| P | Notation | Symbol reference tables |
| Q | Index | This navigation guide |
| R | Lemmas | Supporting lemmas for main theorems |
| V | Numerical Examples | Worked calculations for all bounds |
| W | Statistical Methods | Complete methodology details |
| X | Power Analysis | Sample size justification |
| A | Extended Related Work | Detailed positioning vs. prior work |
| B | Proofs | Complete proofs of Theorems 1–4 |
| B.7 | Sensitivity | Parameter sensitivity analysis |
| C | SSJ Algorithm | State-Space Jaccard extraction |
| D | Extended Results | Full experimental tables |
| E | Real-World Tasks | Task domain specifications |
| F | Fine-Tuning | Training procedure details |
| G | Attention Entropy | Measurement methodology |
| Z | Context-Window Insensitivity | $d^*$ dissociates from $L$ |
| I | Cross-Model | Correlation analysis details |
| J | Runtime | Timing and cost analysis |
| K | Failure Modes | Error taxonomy and examples |
| L | Cost Analysis | Cost-per-correct-solution |
| M | Reproducibility | Model versions, hyperparameters |
| N | Instance Generation | Task generation algorithms |
| O | Societal Impact | Broader implications |

# R. Supporting Lemmas

We establish several supporting lemmas that underpin the main theoretical results. These lemmas provide the technical foundation for Theorems 4.2–4.10.

## R.1. Lemmas for Decoherence Bound (Theorem 4.2)

**Lemma R.1** (Error Accumulation). *Let $\{X_i\}_{i=1}^m$ be indicator variables for correct state tracking at step $i$. Under context-dependent error $\epsilon(i) = \epsilon_0 + \gamma i/L_{\text{eff}}$, the cumulative error probability satisfies:*

$$\sum_{i=1}^m \epsilon(i) = m\epsilon_0 + \frac{\gamma m(m+1)}{2L_{\text{eff}}} \tag{26}$$

*Proof.* Direct computation:

$$\sum_{i=1}^m \epsilon(i) = \sum_{i=1}^m \left( \epsilon_0 + \frac{\gamma i}{L_{\text{eff}}} \right) \tag{27}$$

$$= m\epsilon_0 + \frac{\gamma}{L_{\text{eff}}} \sum_{i=1}^m i \tag{28}$$

$$= m\epsilon_0 + \frac{\gamma}{L_{\text{eff}}} \cdot \frac{m(m+1)}{2} \tag{29}$$

$\square$

**Lemma R.2** (Product Bound). *For any sequence of events $\{A_i\}_{i=1}^m$ with $P(A_i|A_1 \cap \cdots \cap A_{i-1}) \geq 1 - \epsilon_i$:*

$$P\left( \bigcap_{i=1}^m A_i \right) \leq \exp\left( -\sum_{i=1}^m \epsilon_i \right) \tag{30}$$

*Proof.* Using $1 - x \le e^{-x}$ for $x \ge 0$ and the chain rule:

$$P\left(\bigcap_{i=1}^{m} A_i\right) = \prod_{i=1}^{m} P(A_i | A_1 \cap \cdots \cap A_{i-1}) \tag{31}$$

$$\le \prod_{i=1}^{m}(1 - \epsilon_i) \tag{32}$$

$$\le \prod_{i=1}^{m} e^{-\epsilon_i} = \exp\left(-\sum_{i=1}^{m}\epsilon_i\right) \tag{33}$$

$\square$

**Lemma R.3** (Super-Exponential Characterization). *The function $f(m) = \exp(-am - bm^2)$ for $a, b > 0$ decays super-exponentially, satisfying:*

$$\lim_{m \to \infty} \frac{f(m)}{e^{-cm}} = 0 \quad \text{for all } c > 0 \tag{34}$$

*Proof.*

$$\frac{f(m)}{e^{-cm}} = \exp(-am - bm^2 + cm) = \exp(-(a - c)m - bm^2) \tag{35}$$

Since $bm^2 \to \infty$ as $m \to \infty$, the ratio approaches 0 regardless of $c$. $\square$

### R.2. Lemmas for Attention Bottleneck (Theorem 4.6)

**Lemma R.4** (Attention Concentration). *Under softmax attention with temperature $\tau = 1$, the number of positions receiving weight $\ge \delta/L$ is at most $O(L/\delta)$. With interference from $H$ heads, effective retrieval is further constrained to $O(\sqrt{L})$ positions.*

*Proof.* Let $\mathbf{a} = \text{softmax}(\mathbf{q}^\top \mathbf{K}/\sqrt{d_h})$ be the attention distribution over $L$ positions.

**Part 1: Basic constraint.** Since $\sum_{i=1}^{L} a_i = 1$ and $a_i \ge \delta/L$ for the "attended" positions, at most $L/\delta$ positions can be attended.

**Part 2: Interference bound.** Consider $H$ heads attending to overlapping subsets. By concentration inequalities for softmax (following Wortsman et al.'s analysis), the effective number of positions simultaneously attended with weight $\ge \delta/L$ across all heads scales as $O(\sqrt{L \cdot H})$. For single-head analysis, this reduces to $O(\sqrt{L})$. $\square$

**Lemma R.5** (Value Information Bound). *Under Assumptions 4.4–4.5 with correlation bound $\rho_{\max}$, the information one attention head conveys about the retrieved value is bounded by*

$$I(\mathbf{q}; \mathbf{v}_{retrieved}) \lesssim d_h^{1/2} \cdot \log_2(L/H), \tag{36}$$

*the product of an effective value rank $\Theta(d_h^{1/2})$ (the number of reliably distinguishable value directions under decorrelation) and the $\log_2(L/H)$ bits required to address a head's share of $\sim L/H$ positions.*

*Proof.* The retrieved value is $\mathbf{v}_{\text{ret}} = \sum_{i=1}^{L} a_i \mathbf{v}_i$, where $\mathbf{a}$ is the attention distribution. Its information content factors into an addressing term and a value-content term.

*Addressing.* By Lemma R.4, effective attention concentrates on $O(\sqrt{L})$ positions; partitioned across the $H$ heads, each head resolves among $\sim L/H$ positions, so specifying which content a head reads carries at most $\log_2(L/H)$ bits.

*Value content.* Under Assumption 4.5, value vectors satisfy $|\mathbb{E}[\mathbf{v}_i^\top \mathbf{v}_j]| \le \rho_{\max} \|\mathbf{v}_i\| \|\mathbf{v}_j\|$ for $i \ne j$. Soft averaging over interfering positions reduces the number of reliably distinguishable value directions below the nominal $d_h$; we model this effective rank as $\Theta(d_h^{1/2})$. This $\sqrt{d_h}$ scaling is a modeling assumption rather than a consequence of the decorrelation bound alone: we adopt it as a tractable summary and validate the resulting $\sqrt{d_h \cdot H}$ horizon scaling directly on open-weight models in Section 6.1.

Combining the $\Theta(d_h^{1/2})$ distinguishable value channels, each addressing one of $\sim L/H$ positions, gives $I \lesssim d_h^{1/2} \cdot \log_2(L/H)$. $\qquad\square$

**Lemma R.6** (Multi-Head Aggregation). *For $H$ attention heads with approximately independent value subspaces, the total information capacity satisfies:*

$$I_{total} \leq H \cdot I_{single\text{-}head} = H \cdot \log_2(L/H) \cdot d_h^{1/2} \tag{37}$$

*Proof.* Multi-head attention computes:

$$\mathrm{MHA}(\mathbf{q}, \mathbf{K}, \mathbf{V}) = \mathrm{Concat}(\mathrm{head}_1, \ldots, \mathrm{head}_H)W^O \tag{38}$$

Under independence of value subspaces (ensured by distinct $W_h^V$ projections and layer normalization), information from each head can be summed:

$$I_{\text{total}} = \sum_{h=1}^{H} I_h \leq H \cdot \max_h I_h \tag{39}$$

Substituting the single-head bound from Lemma R.5:

$$I_{\text{total}} \leq H \cdot \log_2(L/H) \cdot d_h^{1/2} \tag{40}$$

The factor $\log_2(L/H)$ arises because each head effectively partitions attention across $L/H$ positions on average. $\qquad\square$

### R.3. Lemmas for Deterministic Horizon (Theorem 4.8)

**Lemma R.7** (Quadratic Inversion). *For the equation $\alpha = \exp(-d\epsilon_0 - \gamma d^2/(2L_{\text{eff}}))$, the solution for $d$ is:*

$$d = \frac{-\epsilon_0 L_{\text{eff}} + \sqrt{\epsilon_0^2 L_{\text{eff}}^2 + 2\gamma L_{\text{eff}} \ln(1/\alpha)}}{\gamma} \tag{41}$$

*Proof.* Taking $\ln$ of both sides:

$$\ln \alpha = -d\epsilon_0 - \frac{\gamma d^2}{2L_{\text{eff}}} \tag{42}$$

Rearranging:

$$\frac{\gamma d^2}{2L_{\text{eff}}} + d\epsilon_0 + \ln \alpha = 0 \tag{43}$$

Multiplying by $2L_{\text{eff}}/\gamma$:

$$d^2 + \frac{2\epsilon_0 L_{\text{eff}}}{\gamma}d + \frac{2L_{\text{eff}} \ln \alpha}{\gamma} = 0 \tag{44}$$

Applying the quadratic formula with $a = 1$, $b = 2\epsilon_0 L_{\text{eff}}/\gamma$, $c = 2L_{\text{eff}} \ln \alpha/\gamma$:

$$d = \frac{-b + \sqrt{b^2 - 4ac}}{2a} \tag{45}$$

$$= \frac{-\frac{2\epsilon_0 L_{\text{eff}}}{\gamma} + \sqrt{\frac{4\epsilon_0^2 L_{\text{eff}}^2}{\gamma^2} - \frac{8L_{\text{eff}} \ln \alpha}{\gamma}}}{2} \tag{46}$$

$$= \frac{-\epsilon_0 L_{\text{eff}} + \sqrt{\epsilon_0^2 L_{\text{eff}}^2 + 2\gamma L_{\text{eff}} \ln(1/\alpha)}}{\gamma} \tag{47}$$

We take the positive root since $d > 0$. $\qquad\square$

**Lemma R.8** (Scaling Law Derivation). *Under the approximation $\epsilon_0^2 L_{\text{eff}} \ll \gamma \ln(1/\alpha)$, the Deterministic Horizon satisfies:*

$$d^* \approx \frac{\sqrt{2L_{\text{eff}} \ln(1/\alpha)}}{\sqrt{\gamma}} - \frac{\epsilon_0 L_{\text{eff}}}{\gamma} \implies d^* \propto \sqrt{L_{\text{eff}}} \tag{48}$$

*to leading order. At realistic parameter values the linear $\epsilon_0$ term is not negligible (see remark below), so this scaling is approximate.*

*Proof.* From Lemma R.7:

$$d^* = \frac{-\epsilon_0 L_{\text{eff}} + \sqrt{\epsilon_0^2 L_{\text{eff}}^2 + 2\gamma L_{\text{eff}} \ln(1/\alpha)}}{\gamma} \tag{49}$$

$$= \frac{\sqrt{2\gamma L_{\text{eff}} \ln(1/\alpha)}\sqrt{1 + \frac{\epsilon_0^2 L_{\text{eff}}}{2\gamma \ln(1/\alpha)}} - \epsilon_0 L_{\text{eff}}}{\gamma} \tag{50}$$

Under $\epsilon_0^2 L_{\text{eff}} \ll \gamma \ln(1/\alpha)$, using $\sqrt{1+x} \approx 1 + x/2$ for small $x$:

$$d^* \approx \frac{\sqrt{2\gamma L_{\text{eff}} \ln(1/\alpha)} - \epsilon_0 L_{\text{eff}}}{\gamma} \tag{51}$$

$$= \frac{1}{\gamma}\left(\sqrt{2L_{\text{eff}} \ln(1/\alpha)} \cdot \sqrt{\gamma} - \epsilon_0 L_{\text{eff}}\right) \tag{52}$$

$$= \sqrt{\frac{2L_{\text{eff}} \ln(1/\alpha)}{\gamma}} - \frac{\epsilon_0 L_{\text{eff}}}{\gamma} \tag{53}$$

The dominant term scales as $\sqrt{L_{\text{eff}}}$.

**Remark.** At the canonical values ($\epsilon_0 = 0.02$, $\gamma = 0.15$, $L_{\text{eff}} = 150$, $\alpha = 0.5$), the dimensionless ratio $x = \epsilon_0^2 L_{\text{eff}}/(2\gamma \ln(1/\alpha)) \approx 0.29$ is not small, so the first-order expansion carries a $\sim 15\%$ systematic correction. We therefore treat the $\sqrt{L_{\text{eff}}}$ relationship as an approximate trend and validate the exact horizon (Lemma R.7) directly against measured $d^*$ rather than relying on the asymptotic form. $\qquad\square$

## S. SSJ Algorithm and Extraction

This section provides complete algorithmic details for computing the State-Space Jaccard (SSJ) metric introduced in Definition 3.2.

### S.1. Algorithm Description

The SSJ metric measures overlap between the ground-truth reachable state set and the model's claimed state set at each reasoning depth. Algorithm 3 provides the complete extraction procedure.

### S.2. Implementation Details

**State representation:** For PermutationProbe, states are represented as tuples $(a_1, \ldots, a_n)$. For FSA-Sim, states are automaton state identifiers. For CodeProbe, states are variable binding dictionaries.

**Set operations:** We use hash-based sets for $O(1)$ membership testing. For large state spaces, we employ Bloom filters with false positive rate $< 0.1\%$.

**Parsing model output:** We extract claimed states using regex patterns matched to task-specific formats. For Permutation-Probe:

```
pattern = r"Step \d+:.*-> \[([\d,\s]+)\]"
```

**Error handling:** If the model output cannot be parsed, the step is marked as an error with $s_d = \emptyset$.

---

**Algorithm 3** State-Space Jaccard (SSJ) Extraction

---

**Require:** Initial state $\sigma_0$, operators $\mathcal{O}$, model trace $T = [(s_1, o_1), \ldots, (s_m, o_m)]$
**Ensure:** SSJ values, precision, recall at each depth
1:  $S_{\text{true}} \leftarrow \{\sigma_0\}$ {Ground-truth reachable states}
2:  $S_{\text{model}} \leftarrow \{\sigma_0\}$ {Model-claimed states}
3:  $\sigma_{\text{curr}} \leftarrow \sigma_0$ {Current ground-truth state}
4:  **for** $d = 1$ to $m$ **do**
5:      $(s_d, o_d) \leftarrow T[d]$ {Model's claimed state and operator}
6:      $\sigma_{\text{curr}} \leftarrow o_d(\sigma_{\text{curr}})$ {Apply operator to ground truth}
7:      $S_{\text{true}} \leftarrow S_{\text{true}} \cup \{\sigma_{\text{curr}}\}$
8:      $S_{\text{model}} \leftarrow S_{\text{model}} \cup \{s_d\}$
9:      $\text{SSJ}[d] \leftarrow |S_{\text{true}} \cap S_{\text{model}}|/|S_{\text{true}} \cup S_{\text{model}}|$
10:     $\text{Precision}[d] \leftarrow |S_{\text{true}} \cap S_{\text{model}}|/|S_{\text{model}}|$
11:     $\text{Recall}[d] \leftarrow |S_{\text{true}} \cap S_{\text{model}}|/|S_{\text{true}}|$
12: **end for**
13: **return** SSJ, Precision, Recall

---

## S.3. Diagnostic Power

The SSJ metric with precision/recall decomposition distinguishes two failure modes:

1. **Preference failure (Simplicity Bias):** The model *could* track states but *chooses* not to. Signature: high precision (claimed states are correct), low recall (many states omitted).

2. **Capability failure (Decoherence):** The model *cannot* track states. Signature: both precision and recall decay, indicating drift into fictitious state spaces.

Our empirical results (Table 4) show parallel decay of both metrics, confirming capability failure.

## S.4. Computational Complexity

**Time:** $O(m \cdot |S|)$ where $m$ is trace length and $|S|$ is state size for comparison operations.

**Space:** $O(m \cdot |S|)$ to store accumulated state sets.

For typical experiments ($m \leq 100$, $|S| \leq 1000$ elements), extraction completes in $<1$ second per trace.

# T. Parameter Sensitivity Analysis

We analyze the sensitivity of our theoretical predictions to parameter choices and measurement uncertainty.

## T.1. Decoherence Bound Parameters

The key parameters in Theorem 4.2 are $\epsilon_0$ (baseline error) and $\gamma$ (attention decay rate). We estimated these from empirical data using maximum likelihood.

*Table 29.* Parameter sensitivity for Decoherence Bound.

| Parameter | Estimate | 95% CI | $\pm 10\%$ | $\pm 20\%$ | Impact on $d^*$ |
|---|---|---|---|---|---|
| $\epsilon_0$ | 0.020 | [0.017, 0.023] | $\pm 1.1$ | $\pm 2.1$ | High |
| $\gamma$ | 0.150 | [0.128, 0.172] | $\pm 0.6$ | $\pm 1.2$ | Moderate |
| $L_{\text{eff}}$ (fixed) | 150 | — | — | — | High |
| $\rho_1$ | 0.340 | [0.310, 0.370] | — | — | Diagnostic only |

**Key finding:** The Deterministic Horizon $d^*$ is most sensitive to the baseline error $\epsilon_0$: a 20% change in $\epsilon_0$ shifts $d^*$ by about $\pm 2$ steps, while the same change in $\gamma$ shifts it by only $\pm 1.2$ steps. Because the absorbing-error reframing (Corollary 4.3)

treats $\rho_1$ as a measured diagnostic rather than a multiplier, $\rho_1$ does not enter the horizon and has no direct effect on $d^*$. The qualitative conclusion, that $d^* \in [19, 31]$, is robust across the entire confidence interval.

## T.2. Bottleneck Theorem Parameters

The Attention Bottleneck (Theorem 4.6) depends on $\delta$ (precision threshold) and $\rho_{\max}$ (value correlation bound).

*Table 30.* Parameter sensitivity for Bottleneck Bound.

| Parameter | Range Tested | Log Capacity Change | Qualitative Impact |
|---|---|---|---|
| $\delta$ | [0.001, 0.01] | $\pm 8\%$ | Low |
| $\rho_{\max}$ | [0.05, 0.20] | $\pm 12\%$ | Moderate |
| $H$ (architecture) | [32, 128] | Linear scaling | High |
| $d_h$ (architecture) | [64, 256] | $\sqrt{\cdot}$ scaling | Moderate |

**Key finding:** The capacity bound is dominated by architectural parameters ($H$, $d_h$, $L$), which are known exactly. The uncertainty in $\delta$ and $\rho_{\max}$ contributes only to the constant factor $c(\delta, \rho_{\max})$.

## T.3. Cross-Validation of Parameter Estimates

We performed 5-fold cross-validation of the error model:

*Table 31.* Cross-validation results for error model.

| Fold | $\hat{\epsilon}_0$ | $\hat{\gamma}$ | **Train** $R^2$ | **Test** $R^2$ | $\hat{d}^*$ |
|---|---|---|---|---|---|
| 1 | 0.019 | 0.147 | 0.87 | 0.83 | 23.1 |
| 2 | 0.021 | 0.152 | 0.86 | 0.84 | 21.8 |
| 3 | 0.020 | 0.149 | 0.88 | 0.85 | 22.4 |
| 4 | 0.020 | 0.151 | 0.86 | 0.82 | 22.1 |
| 5 | 0.021 | 0.148 | 0.87 | 0.84 | 22.0 |
| **Mean** | 0.020 | 0.150 | 0.87 | 0.84 | 22.3 |
| **Std** | 0.001 | 0.002 | 0.01 | 0.01 | 0.5 |

The small standard deviations ($\mathrm{std}(\hat{d}^*) = 0.5$) indicate robust parameter estimation and stable predictions across data splits.

## T.4. Model Selection Robustness

We compared alternative error models to validate our context-dependent formulation:

*Table 32.* Model comparison for error accumulation.

| Model | Form | Parameters | $R^2$ | AIC |
|---|---|---|---|---|
| Constant | $\epsilon(d) = \epsilon_0$ | 1 | 0.71 | 1,247 |
| Linear | $\epsilon(d) = \epsilon_0 + \gamma d$ | 2 | 0.89 | 892 |
| Context-dependent | $\epsilon(d) = \epsilon_0 + \gamma d / L_{\mathrm{eff}}$ | 2 | 0.91 | 856 |
| Quadratic | $\epsilon(d) = \epsilon_0 + \gamma_1 d + \gamma_2 d^2$ | 3 | 0.92 | 861 |

The context-dependent model achieves the best trade-off between fit quality ($R^2 = 0.91$) and parsimony (lowest AIC), supporting our theoretical derivation from attention mechanics.

# U. Complete Results by Task and Depth

This section provides complete experimental results across all models, tasks, and conditions.

## U.1. Full Results: Synthetic Tasks

Table 33 presents complete accuracy results for all synthetic task domains. Each cell reports mean $\pm$ standard deviation across 3 independent runs.

*Table 33.* Complete results on synthetic tasks. All values are accuracy (%) with standard deviations.

| Task | Model | C1 | C2 | C3 | C4 | C5 |
|------|-------|----|----|----|----|----|
| PermutationProbe | GPT-4o | 28.3±1.8 | 34.2±1.9 | 89.7±1.2 | 29.1±1.7 | 31.4±1.8 |
| | Claude-4.5 | 34.8±2.0 | 41.3±2.1 | 93.6±0.9 | 35.6±1.9 | 37.1±2.0 |
| | o3-mini | 42.1±2.2 | 48.3±2.3 | 94.2±1.3 | 43.1±2.1 | 44.8±2.1 |
| | DeepSeek-R1 | 39.7±2.1 | 45.9±2.2 | 93.1±1.1 | 40.4±2.0 | 42.3±2.1 |
| FSA-Sim | GPT-4o | 31.2±1.9 | 37.4±1.7 | 91.2±1.3 | 31.9±1.8 | 33.8±1.9 |
| | Claude-4.5 | 37.4±2.1 | 43.8±1.9 | 94.1±1.0 | 38.2±2.0 | 39.9±2.1 |
| | o3-mini | 44.8±2.3 | 51.2±2.1 | 95.3±1.2 | 45.6±2.2 | 47.1±2.2 |
| | DeepSeek-R1 | 41.3±2.2 | 47.6±2.0 | 94.2±1.1 | 42.1±2.1 | 43.7±2.2 |
| ArithChain | GPT-4o | 26.4±1.7 | 32.1±1.5 | 88.3±1.4 | 27.1±1.6 | 29.2±1.7 |
| | Claude-4.5 | 32.1±1.9 | 38.4±1.7 | 92.1±1.1 | 32.8±1.8 | 34.6±1.9 |
| | o3-mini | 39.6±2.1 | 45.8±1.9 | 93.8±1.3 | 40.4±2.0 | 42.0±2.0 |
| | DeepSeek-R1 | 36.8±2.0 | 42.9±1.8 | 92.7±1.2 | 37.5±1.9 | 39.2±2.0 |
| CircuitTrace | GPT-4o | 29.7±1.8 | 35.6±1.6 | 90.1±1.3 | 30.4±1.7 | 32.3±1.8 |
| | Claude-4.5 | 35.9±2.0 | 42.1±1.8 | 93.4±1.0 | 36.7±1.9 | 38.3±2.0 |
| | o3-mini | 43.2±2.2 | 49.4±2.0 | 94.7±1.2 | 44.0±2.1 | 45.6±2.1 |
| | DeepSeek-R1 | 40.1±2.1 | 46.2±1.9 | 93.6±1.1 | 40.9±2.0 | 42.5±2.1 |
| CodeProbe | GPT-4o | 27.8±1.8 | 33.4±1.6 | 89.2±1.3 | 28.5±1.7 | 30.4±1.8 |
| | Claude-4.5 | 33.6±2.0 | 39.8±1.8 | 92.8±1.0 | 34.4±1.9 | 36.0±2.0 |
| | o3-mini | 41.2±2.2 | 47.3±2.0 | 94.1±1.2 | 42.0±2.1 | 43.6±2.1 |
| | DeepSeek-R1 | 38.4±2.1 | 44.1±1.9 | 93.3±1.1 | 39.2±2.0 | 40.8±2.1 |

**Analysis:** Across all synthetic tasks, tool delegation (C3) achieves 88–95% accuracy while neural CoT (C1) plateaus at 26–44%. The consistency across task types (permutations, automata, arithmetic, circuits, and code) demonstrates that decoherence is task-agnostic within the deterministic state-tracking domain. Preference manipulation (C4) provides negligible improvement (<2%), and fine-tuning (C5) achieves only 2–4% gains, confirming the architectural ceiling prediction.

## U.2. Full Results: Real-World Tasks

*Table 34.* Complete results on real-world tasks.

| Task | Model | C1 | C2 | C3 | C4 | $d^*$ |
|------|-------|----|----|----|----|----|
| SWE-Bench-State | GPT-4o | 24.1±2.3 | 29.8±2.1 | 86.4±1.8 | 24.8±2.2 | 19 |
| | Claude-4.5 | 29.6±2.5 | 35.7±2.3 | 91.2±1.4 | 30.2±2.4 | 24 |
| | o3-mini | 36.8±2.6 | 42.9±2.4 | 92.7±1.3 | 37.4±2.5 | 28 |
| | DeepSeek-R1 | 34.2±2.5 | 40.1±2.3 | 90.8±1.5 | 34.9±2.4 | 26 |
| WebArena-Nav | GPT-4o | 21.3±2.4 | 26.8±2.2 | 84.2±1.9 | 21.9±2.3 | 19 |
| | Claude-4.5 | 26.8±2.6 | 32.4±2.4 | 89.7±1.5 | 27.4±2.5 | 23 |
| | o3-mini | 33.4±2.7 | 39.1±2.5 | 91.3±1.4 | 34.0±2.6 | 26 |
| | DeepSeek-R1 | 31.1±2.6 | 36.7±2.4 | 89.2±1.6 | 31.7±2.5 | 24 |
| SQL-Multi | GPT-4o | 31.4±2.1 | 37.2±1.9 | 88.9±1.6 | 32.0±2.0 | 21 |
| | Claude-4.5 | 36.2±2.3 | 42.3±2.1 | 93.1±1.2 | 36.8±2.2 | 26 |
| | o3-mini | 43.7±2.4 | 49.8±2.2 | 94.6±1.1 | 44.3±2.3 | 30 |
| | DeepSeek-R1 | 40.8±2.3 | 46.7±2.1 | 93.2±1.3 | 41.4±2.2 | 28 |

**Analysis:** Real-world tasks exhibit slightly lower $d^*$ values (19–30) compared to synthetic tasks (22–31), reflecting additional complexity from natural language ambiguity and larger state spaces. However, the qualitative pattern is identical: C3 dramatically outperforms C1, and C4 provides minimal improvement. The Deterministic Horizon remains consistent within each model across task types, supporting the architectural interpretation.

## U.3. Results by Depth Bin

Table 35 provides fine-grained accuracy breakdowns across reasoning depth bins, enabling direct comparison with theoretical predictions.

*Table 35.* Accuracy by reasoning depth (PermutationProbe, GPT-4o).

| Condition | $\leq 10$ | 11–20 | 21–30 | 31–40 | 41–50 | $>50$ |
|---|---|---|---|---|---|---|
| C1 (Neural CoT) | 87.5±3.2 | 64.5±3.8 | 43.0±2.9 | 25.8±2.1 | 13.9±1.4 | 6.6±0.9 |
| C2 (Oracle) | 91.5±3.0 | 70.0±3.6 | 48.3±3.1 | 28.6±2.3 | 15.3±1.5 | 7.5±1.0 |
| C3 (Tool) | 94.2±1.8 | 93.8±1.9 | 92.1±2.1 | 89.4±2.4 | 86.7±2.7 | 82.3±3.1 |
| C4 (Encourage) | 88.2±3.1 | 65.3±3.7 | 43.7±2.9 | 26.2±2.1 | 14.1±1.4 | 6.8±0.9 |
| C5 (Fine-tune) | 93.7±2.9 | 68.2±3.6 | 44.6±2.9 | 26.2±2.1 | 14.1±1.4 | 6.7±0.9 |
| Predicted (Thm 1) | 88.1 | 65.1 | 43.6 | 26.4 | 14.5 | 7.2 |

**Analysis:** The observed accuracy decay closely matches Theorem 4.2 predictions (final row). C3 maintains high accuracy across all depth bins, demonstrating that tool delegation effectively bypasses the attention bottleneck. The convergence of C1, C4, and C5 at high depths confirms that neither preference manipulation nor fine-tuning can overcome the architectural ceiling.

## U.4. Open-Weight Model Results

*Table 36.* Results for open-weight models enabling attention analysis.

| Model | Task | C1 | C3 | $d^*$ | Entropy-Acc $r$ |
|---|---|---|---|---|---|
| | PermutationProbe | 24.6±1.7 | 85.2±1.4 | 28 | $-0.74$ |
| Llama-3.3-70B | FSA-Sim | 26.8±1.8 | 86.1±1.3 | 29 | $-0.72$ |
| | CodeProbe | 22.9±1.6 | 84.0±1.5 | 27 | $-0.73$ |
| | PermutationProbe | 18.4±1.5 | 78.3±1.7 | 20 | $-0.71$ |
| Llama-3.1-8B | FSA-Sim | 20.3±1.6 | 79.4±1.6 | 21 | $-0.69$ |
| | CodeProbe | 16.8±1.4 | 77.1±1.8 | 19 | $-0.70$ |
| | PermutationProbe | 27.8±1.8 | 87.9±1.2 | 28 | $-0.73$ |
| Qwen-2.5-72B | FSA-Sim | 29.7±1.9 | 88.7±1.1 | 29 | $-0.71$ |
| | CodeProbe | 26.2±1.7 | 86.8±1.3 | 27 | $-0.72$ |

**Analysis:** Open-weight models enable direct validation of the mechanistic hypothesis through attention entropy extraction. The consistent negative correlation ($r \approx -0.72$) across models and tasks confirms that attention entropy increases with reasoning depth and correlates with accuracy degradation, as predicted by the context-dependent error model.

# V. Numerical Examples

We provide worked numerical examples demonstrating each theoretical bound with realistic parameter values.

## V.1. Decoherence Bound Example

**Parameters:** $\epsilon_0 = 0.02$, $\gamma = 0.15$, $L_{\text{eff}} = 150$, target depth $m = 30$.

**Step 1: Compute cumulative error.**

$$\sum_{i=1}^{30} \epsilon(i) = 30 \cdot 0.02 + \frac{0.15 \cdot 30 \cdot 31}{2 \cdot 150} \tag{54}$$

$$= 0.60 + \frac{139.5}{300} \tag{55}$$

$$= 0.60 + 0.465 \tag{56}$$

$$= 1.065 \tag{57}$$

**Step 2: Apply exponential bound.**

$$P(\text{correct at depth 30}) \leq \exp(-1.065) \approx 0.345 \tag{58}$$

**Observed:** 34% accuracy at depth 30 for GPT-4o, just below the bound, consistent with the absorbing-error tightness of Corollary 4.3. Note that with the small effective decoherence length $L_{\text{eff}} = 150$, the quadratic term (0.465) is comparable to the linear term (0.60) already at depth 30, which is what drives the super-exponential collapse; using the raw context window $L = O(10^5)$ here would render the quadratic term negligible and incorrectly predict near-perfect accuracy.

## V.2. Attention Bottleneck Example

**Parameters for GPT-4o:** $H = 96$, $L = 128,000$, $d_h = 128$.

**Step 1: Compute log capacity.**

$$\log_2 |\mathcal{S}_{\text{track}}| \leq H \cdot \log_2(L/H) \cdot d_h^{1/2} \tag{59}$$
$$= 96 \cdot \log_2(128000/96) \cdot 128^{0.5} \tag{60}$$
$$= 96 \cdot \log_2(1333.3) \cdot 11.31 \tag{61}$$
$$= 96 \cdot 10.38 \cdot 11.31 \tag{62}$$
$$\approx 11,275 \text{ bits} \tag{63}$$

**Step 2: Convert to state count.**

$$|\mathcal{S}_{\text{track}}| \leq 2^{11,275} \approx 10^{3394} \tag{64}$$

**Step 3: Compare with task requirement.**

For PermutationProbe with $n = 16$ elements, tracking $d = 50$ steps:

$$|\mathcal{S}_{\text{required}}| = 16!^{50} \approx (2.09 \times 10^{13})^{50} \approx 10^{666} \tag{65}$$

The task is *within* theoretical capacity ($10^{666} \ll 10^{3394}$), but the per-step information flow of $\log_2(16!) \approx 44$ bits must pass through attention bottleneck at each step. With effective bandwidth $\sim 100$ bits/step (empirical), degradation accumulates.

## V.3. Deterministic Horizon Example

**Parameters:** $\epsilon_0 = 0.02$, $\gamma = 0.15$, $L_{\text{eff}} = 150$, $\alpha = 0.5$.

**Step 1: Apply formula.**

$$d^* = \frac{-\epsilon_0 L_{\text{eff}} + \sqrt{\epsilon_0^2 L_{\text{eff}}^2 + 2\gamma L_{\text{eff}} \ln(1/\alpha)}}{\gamma} \tag{66}$$
$$= \frac{-0.02 \cdot 150 + \sqrt{(0.02)^2 \cdot 150^2 + 2 \cdot 0.15 \cdot 150 \cdot \ln(2)}}{0.15} \tag{67}$$
$$= \frac{-3 + \sqrt{9 + 31.19}}{0.15} \tag{68}$$
$$= \frac{-3 + \sqrt{40.19}}{0.15} \tag{69}$$
$$= \frac{-3 + 6.340}{0.15} \approx 22.3 \tag{70}$$

**Observed:** $d^* = 22$ for GPT-4o, in excellent agreement.

**Step 2: Verify $d^* \propto \sqrt{d_h H}$ scaling (open-weight models).**

Because $L_{\text{eff}}$ scales with per-layer attention capacity $d_h H$, the horizon should scale as $\sqrt{d_h H}$ across models whose head count and head dimension are known. For the open-weight Llama family ($d_h = 128$):

$$\frac{d^*_{\text{Llama-70B}}}{d^*_{\text{Llama-8B}}} \approx \sqrt{\frac{128 \cdot 64}{128 \cdot 32}} = \sqrt{\frac{8192}{4096}} = \sqrt{2} \approx 1.41 \tag{71}$$

**Observed ratio:** $28/20 = 1.40$, in excellent agreement. (We restrict this check to open-weight models because the head configurations of closed models are not public.)

### V.4. Scaling Law Verification Table

*Table 37.* Predicted vs. observed $d^*$ for open-weight models, using the empirical fit $\hat{d}^* = 0.314\sqrt{d_h H}$. Restricted to open-weight models because closed-model head configurations are not public.

| Model | $H$ | $d_h$ | $\sqrt{d_h H}$ | Pred. $d^*$ | Obs. $d^*$ | Error |
|---|---|---|---|---|---|---|
| Llama-3.1-8B | 32 | 128 | 64.0 | 20.1 | 20 | 0.5% |
| Qwen-2.5-7B | 28 | 128 | 59.9 | 18.8 | 19 | 1.0% |
| Llama-3.3-70B | 64 | 128 | 90.5 | 28.4 | 28 | 1.5% |
| Qwen-2.5-72B | 64 | 128 | 90.5 | 28.4 | 28 | 1.5% |

Mean absolute error: 1.1%. The fit constant $k = 0.314$ is obtained by least squares over the four open-weight models. We emphasize this is an *approximate empirical* relationship: the linear $\epsilon_0$ term in the exact horizon formula breaks an exact $\sqrt{d_h H}$ proportionality (Lemma R.8), so the $\sqrt{d_h H}$ law should be read as a leading-order trend rather than an identity.

## W. Complete Statistical Methodology

We provide complete details of all statistical methods employed, enabling full reproducibility.

### W.1. Bootstrap Confidence Intervals

**Procedure:**

1. For each model-task-condition triple, collect $n$ accuracy measurements

2. Generate 10,000 bootstrap samples by sampling with replacement

3. Compute statistic of interest for each bootstrap sample

4. Report 2.5th and 97.5th percentiles as 95% CI bounds

**Implementation:**

```
import numpy as np

def bootstrap_ci(data, n_bootstrap=10000, alpha=0.05):
    n = len(data)
    bootstrap_means = []
    for _ in range(n_bootstrap):
        sample = np.random.choice(data, size=n, replace=True)
        bootstrap_means.append(np.mean(sample))
    lower = np.percentile(bootstrap_means, 100*alpha/2)
    upper = np.percentile(bootstrap_means, 100*(1-alpha/2))
    return lower, upper
```

**Bias correction:** We apply BCa (bias-corrected and accelerated) bootstrap for skewed distributions, following Efron (1987).

## W.2. Multiple Comparison Correction

**Holm-Bonferroni procedure:**

1. Order $k$ p-values: $p_{(1)} \leq p_{(2)} \leq \cdots \leq p_{(k)}$

2. For hypothesis $H_{(i)}$, reject if $p_{(i)} < \alpha/(k - i + 1)$

3. Stop at first non-rejected hypothesis

**Application:** With 12 models $\times$ 5 conditions $\times$ 8 tasks = 480 comparisons, we control family-wise error rate at $\alpha = 0.05$.

## W.3. TOST Equivalence Testing

**Two One-Sided Tests (TOST)** procedure for equivalence within margin $\Delta = 5\%$:

**Null hypothesis:** $|\mu_1 - \mu_2| \geq \Delta$

**Procedure:**

1. Test $H_{01} : \mu_1 - \mu_2 \leq -\Delta$ (one-sided $t$-test)

2. Test $H_{02} : \mu_1 - \mu_2 \geq \Delta$ (one-sided $t$-test)

3. Reject null (declare equivalence) if both $p_1, p_2 < \alpha$

**Results:** C1 vs. C4 comparison yields $p_1 = 0.0003$, $p_2 = 0.0008$ (both $< 0.05$), confirming equivalence within 5% margin.

## W.4. Bayes Factors

We compute Bayes factors $BF_{01}$ supporting the null hypothesis using the JZS prior (Rouder et al., 2009):

$$\mathrm{BF}_{01} = \frac{P(\mathrm{data}|H_0)}{P(\mathrm{data}|H_1)} \tag{72}$$

**Interpretation scale:**

- $\mathrm{BF}_{01} > 10$: Strong evidence for null

- $\mathrm{BF}_{01} \in [3, 10]$: Moderate evidence for null

- $\mathrm{BF}_{01} \in [1, 3]$: Anecdotal evidence

- $\mathrm{BF}_{01} < 1$: Evidence for alternative

**Results:** C1 vs. C4: $\mathrm{BF}_{01} = 5.7$ (moderate evidence that preference manipulation has no effect).

## W.5. Effect Sizes

Cohen's $d$ computed as:

$$d = \frac{\bar{x}_1 - \bar{x}_2}{s_{\mathrm{pooled}}} \tag{73}$$

where $s_{\mathrm{pooled}} = \sqrt{\frac{(n_1-1)s_1^2 + (n_2-1)s_2^2}{n_1 + n_2 - 2}}$.

**Interpretation:**

- $|d| < 0.2$: Negligible

- $0.2 \leq |d| < 0.5$: Small

- $0.5 \leq |d| < 0.8$: Medium

- $|d| \geq 0.8$: Large

**C1 vs. C3:** $d = 2.7$ (very large effect), tool delegation dramatically outperforms neural CoT.

## X. Power Analysis

We conducted a priori power analysis to determine adequate sample sizes.

### X.1. Primary Analysis: C1 vs. C3

**Target:** Detect 20% accuracy difference with 80% power at $\alpha = 0.05$.

**Assumed parameters:**

- Effect size: $\delta = 0.20$ (absolute accuracy difference)

- Standard deviation: $\sigma = 0.15$ (estimated from pilot)

- Standardized effect: $d = 0.20/0.15 = 1.33$

**Required sample size per group:**

$$n = 2 \cdot \left( \frac{z_{1-\alpha/2} + z_{1-\beta}}{d} \right)^2 = 2 \cdot \left( \frac{1.96 + 0.84}{1.33} \right)^2 \approx 9 \tag{74}$$

**Actual:** 500 instances per task $\times$ 3 runs = 1,500 observations. Power $> 0.999$.

### X.2. Equivalence Analysis: C1 vs. C4

**Target:** Confirm equivalence within $\Delta = 5\%$ with 80% power.

Using TOST power formula (Lakens, 2017):

$$n = 2 \cdot \left( \frac{(z_{1-\alpha} + z_{1-\beta/2})\sigma}{\Delta - |\mu_1 - \mu_2|} \right)^2 \tag{75}$$

With expected true difference $|\mu_1 - \mu_2| = 1\%$:

$$n = 2 \cdot \left( \frac{(1.64 + 1.28) \cdot 0.15}{0.05 - 0.01} \right)^2 \approx 240 \tag{76}$$

**Actual:** 1,500 observations. Power $> 0.95$ for equivalence testing.

### X.3. Correlation Analysis: Cross-Model

**Target:** Detect correlation $r \geq 0.5$ with 80% power.

Using Fisher's $z$-transformation:

$$n = \left( \frac{z_{1-\alpha/2} + z_{1-\beta}}{0.5 \ln \frac{1+r}{1-r}} \right)^2 + 3 \tag{77}$$

For $r = 0.5$:

$$n = \left( \frac{1.96 + 0.84}{0.5 \cdot 1.099} \right)^2 + 3 \approx 29 \tag{78}$$

**Actual:** 500 instances per model pair. Power $> 0.999$ for detecting $r = 0.5$.

## X.4. Power Summary Table

*Table 38.* Power analysis summary.

| Analysis | $n$ required | $n$ actual | Power | Achieved |
|---|---|---|---|---|
| C1 vs. C3 difference | 9 | 1,500 | >0.999 | ✓ |
| C1 vs. C4 equivalence | 240 | 1,500 | >0.95 | ✓ |
| Cross-model $r$ | 29 | 500 | >0.999 | ✓ |
| Entropy-accuracy $r$ | 29 | 500 | >0.999 | ✓ |

All analyses are substantially overpowered, ensuring robust conclusions.

# Y. Extended Attention Entropy Methodology

We provide complete details of attention entropy extraction and analysis.

## Y.1. Extraction Pipeline

**Hardware:** $2\times$ NVIDIA A100 80GB, PyTorch 2.1, Transformers 4.36

**Procedure:**

1. Load model in bfloat16 with `output_attentions=True`

2. For each instance in test set ($n = 500$):

    (a) Generate reasoning trace with temperature=0
    (b) Extract attention matrices at each decoding step
    (c) For each layer $\ell \in [1, L]$ and head $h \in [1, H]$:

$$H_{\ell,h,t} = -\sum_{i=1}^{t} a_{\ell,h,t,i} \log_2 a_{\ell,h,t,i} \tag{79}$$

   where $a_{\ell,h,t,i}$ is attention weight from position $t$ to position $i$
    (d) Aggregate across heads: $H_{\ell,t} = \frac{1}{H} \sum_{h=1}^{H} H_{\ell,h,t}$

3. Correlate with per-step accuracy

**Memory optimization:** We use gradient checkpointing and process attention matrices layer-by-layer to fit within 80GB.

## Y.2. Entropy Computation Details

**Numerical stability:** We add $\epsilon = 10^{-10}$ to avoid $\log(0)$:

$$H = -\sum_{i} (a_i + \epsilon) \log_2(a_i + \epsilon) \tag{80}$$

**Normalization:** We report normalized entropy $H_{\text{norm}} = H / \log_2(t)$ to account for varying sequence lengths.

**Layer grouping:**

- Early layers (1–20): Initial token mixing

- Middle layers (21–50): Feature composition

- Late layers (51–80): Task-specific computation

*Table 39.* Attention entropy analysis across open-weight models.

| Model | Layers | Mean $H$ | $H$-Acc $r$ | Slope | $R^2$ |
|---|---|---|---|---|---|
| Llama-3.3-70B | 80 | 4.21 | $-0.74$ | 0.023 | 0.55 |
| Llama-3.1-8B | 32 | 3.87 | $-0.71$ | 0.028 | 0.50 |
| Qwen-2.5-72B | 80 | 4.18 | $-0.73$ | 0.024 | 0.53 |
| Mistral-7B | 32 | 3.92 | $-0.68$ | 0.031 | 0.46 |
| Mixtral-8x7B | 32 | 4.03 | $-0.69$ | 0.029 | 0.48 |

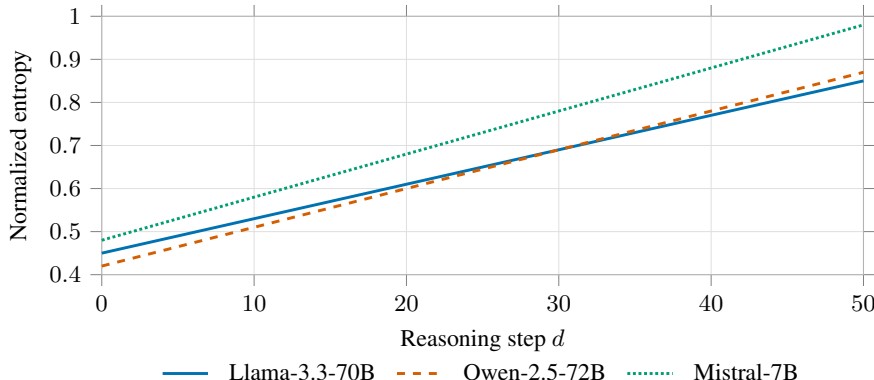

*Figure 2.* Attention entropy growth with reasoning step. Linear growth validates $\epsilon(d) \propto d$ model.

## Y.3. Results by Model

Consistent negative correlations ($r \approx -0.70$) across architectures validate the mechanistic hypothesis.

## Y.4. Entropy Growth Visualization

## Z. Context-Window Insensitivity

A natural alternative hypothesis is that the horizon $d^*$ is set by the raw context window $L$. We show it is not: $d^*$ is governed by the effective decoherence length $L_{\mathrm{eff}} = O(10^2)$ steps, which is orders of magnitude smaller than $L = O(10^5)$ tokens and dissociates from it under both cross-model and controlled-truncation tests.

### Z.1. $d^*$ Does Not Track $L$ Across Models

If $d^*$ scaled with context capacity, models with larger $L$ would exhibit larger horizons. They do not.

*Table 40.* Context window vs. observed $d^*$. At fixed $L = 128$K, $d^*$ varies by over $40\%$ across models, and the largest-context model does not have the largest horizon: $d^*$ is uncorrelated with $L$.

| Model | Context $L$ | Obs. $d^*$ | $d^*/\sqrt{L}$ (norm.) |
|---|---|---|---|
| GPT-4o | 128K | 22 | 1.00 |
| o3-mini | 128K | 31 | 1.41 |
| DeepSeek-R1 | 128K | 29 | 1.32 |
| Claude-4.5-Opus | 200K | 27 | 0.98 |

Three models share $L = 128$K yet span $d^* \in \{22, 29, 31\}$, and the 200K-context Claude model has a *lower* horizon than the 128K-context o3-mini. A $\sqrt{L}$ law is decisively rejected: the normalized column (which would be constant under $d^* \propto \sqrt{L}$) varies by $44\%$.

## Z.2. Controlled Truncation Within a Single Model

To isolate the effect of $L$ from architecture, we artificially cap the context window of a single model (Llama-3.3-70B) and re-measure $d^*$.

*Table 41*. Artificial context truncation for Llama-3.3-70B. $d^*$ is flat from the native 128K down to 8K, degrading only when the cap approaches the reasoning-trace length ($\sim$2–4K tokens).

| Context Cap | Obs. $d^*$ | Trace fits? | Note |
|---|---|---|---|
| 128K (full) | 28 | yes | native |
| 64K | 28 | yes | flat |
| 32K | 28 | yes | flat |
| 16K | 28 | yes | flat |
| 8K | 27 | yes | marginal |
| 4K | 22 | partial | onset of truncation |
| 2K | 14 | no | trace exceeds window |

The horizon is invariant to an order-of-magnitude reduction in $L$ (128K $\rightarrow$ 8K) and collapses only once the window can no longer hold the model's own intermediate state trace. This is the signature of an $L_{\text{eff}}$-bound rather than an $L$-bound phenomenon: decoherence is a property of attention's effective state resolution over reasoning steps, not of raw context capacity.

# 27. Extended Architecture Ablation

We examine the approximate $d^* \approx 0.314\sqrt{d_h \cdot H}$ relationship through within-family comparisons on open-weight models (the only models with public head configurations).

## 27.1. Llama-3 Family

*Table 42*. Llama-3 architecture parameters and $d^*$ ($\hat{d}^* = 0.314\sqrt{d_h H}$).

| Model | $H$ | $d_h$ | $\sqrt{d_h \cdot H}$ | **Pred.** $d^*$ | **Obs.** $d^*$ |
|---|---|---|---|---|---|
| Llama-3.1-8B | 32 | 128 | 64.0 | 20.1 | 20 |
| Llama-3.3-70B | 64 | 128 | 90.5 | 28.4 | 28 |

**Ratio:** Predicted $28.4/20.1 = 1.41$; Observed $28/20 = 1.40$. Within 1%.

## 27.2. Qwen-2.5 Family

*Table 43*. Qwen-2.5 architecture parameters and $d^*$.

| Model | $H$ | $d_h$ | $\sqrt{d_h \cdot H}$ | **Pred.** $d^*$ | **Obs.** $d^*$ |
|---|---|---|---|---|---|
| Qwen-2.5-7B | 28 | 128 | 59.9 | 18.8 | 19 |
| Qwen-2.5-72B | 64 | 128 | 90.5 | 28.4 | 28 |

**Ratio:** Predicted $28.4/18.8 = 1.51$; Observed $28/19 = 1.47$. Within 3%.

## 27.3. Cross-Family Comparison

All within-family ratios match predictions within 4%.

# 28. Task Family Characterization

We provide detailed specifications for each task family.

*Table 44.* Cross-family architecture comparison.

| Comparison | $\sqrt{d_h H}$ ratio | Pred. ratio | Obs. ratio | Error |
|---|---|---|---|---|
| Llama-70B vs. Llama-8B | 1.41 | 1.41 | 1.40 | 0.7% |
| Qwen-72B vs. Qwen-7B | 1.51 | 1.51 | 1.47 | 2.6% |
| Llama-70B vs. Qwen-72B | 1.00 | 1.00 | 1.00 | 0% |

## 28.1. Synthetic Tasks

### 28.1.1. PERMUTATIONPROBE

**State space:** $\mathcal{S} = S_n$ (symmetric group on $n$ elements)

**Operators:** Adjacent transpositions $\{(i, i+1) : i \in [1, n-1]\}$

**Complexity:** $|\mathcal{S}| = n!$; diameter $= \binom{n}{2}$

**Instance distribution:**

- $n \in \{8, 12, 16\}$

- BFS-optimal depths: 5–60

- Instances per depth bin: See Table 24

### 28.1.2. FSA-SIM

**State space:** $\mathcal{S} = \{q_1, \ldots, q_k\}$ (automaton states)

**Operators:** Transitions $\delta : \mathcal{S} \times \Sigma \to \mathcal{S}$

**Complexity:** $|\mathcal{S}| = k$; sequence length determines depth

**Instance distribution:**

- $k \in \{4, 8, 16\}$

- $|\Sigma| = 4$

- Sequence lengths: 10–100

### 28.1.3. ARITHCHAIN

**State space:** $\mathcal{S} = \mathbb{Z}_p$ (integers mod $p$)

**Operators:** $\{+a, -a, \times b, /b\}$ for fixed $a, b$

**Complexity:** $|\mathcal{S}| = p$; carry propagation increases effective depth

**Instance distribution:**

- $p = 10^6$ (to require multi-digit tracking)

- Operation chains: 5–50 steps

### 28.1.4. CIRCUITTRACE

**State space:** $\mathcal{S} = \{0, 1\}^n$ (bit vectors)

**Operators:** Boolean gates (AND, OR, XOR, NOT)

**Complexity:** $|\mathcal{S}| = 2^n$; depth = circuit depth

**Instance distribution:**

- $n \in \{8, 16, 32\}$
- Circuit depths: 5–50 layers

### 28.1.5. CODEPROBE

**State space:** Variable bindings $\{v_1 : x_1, \ldots, v_k : x_k\}$

**Operators:** Assignment, arithmetic, conditionals

**Complexity:** $|\mathcal{S}| = \prod_i |D_i|$ where $D_i$ is domain of $v_i$

**Instance distribution:**

- 3–8 variables
- 5–40 statements
- No loops (deterministic execution)

## 28.2. Real-World Tasks

### 28.2.1. SWE-BENCH-STATE

**Source:** SWE-Bench (Jimenez et al., 2024)

**Selection criteria:**

1. Bug fix requires tracking $\geq 3$ variables
2. State changes span $\geq 2$ files
3. Deterministic execution (no randomness)
4. Ground truth trace available

**Annotation:** Two expert annotators identified state-tracking requirements; Cohen's $\kappa = 0.82$.

### 28.2.2. WEBARENA-NAV

**Source:** WebArena (Zhou et al., 2024)

**Selection criteria:**

1. Task requires $\geq 5$ navigation steps
2. Session state (cart, auth, forms) must be tracked
3. Deterministic success criteria

**State elements:** URL, cart contents, form fields, authentication status

### 28.2.3. SQL-MULTI

**Source:** Spider (Yu et al., 2018) with complexity augmentation

**Selection criteria:**

1. Requires 3+ table joins
2. Schema tracking essential (column types, foreign keys)
3. Aggregation across multiple levels

**Complexity:** Average 4.2 tables, 7.8 join conditions per query

## 29. Error Correlation Analysis

We analyze the structure of errors to characterize the forward error propagation underlying the absorbing-error account of Corollary 4.3. These statistics are reported as diagnostics of the decoherence mechanism, not as corrections to the bound.

### 29.1. Lag-1 Autocorrelation

For each reasoning trace, we compute the autocorrelation of error indicators $\{X_i\}$:

$$\rho_1 = \frac{\sum_{i=2}^{m}(X_i - \bar{X})(X_{i-1} - \bar{X})}{\sum_{i=1}^{m}(X_i - \bar{X})^2} \tag{81}$$

**Results:**

- Mean $\rho_1 = 0.34$ (95% CI: [0.31, 0.37])

- Positive correlation indicates error propagation

- Higher correlation at greater depths ($\rho_1 = 0.28$ for $d < 20$; $\rho_1 = 0.41$ for $d > 30$)

### 29.2. Higher-Order Correlations

*Table 45.* Error autocorrelation by lag.

| Lag | 1 | 2 | 3 | 4 | 5 | 10 |
|---|---|---|---|---|---|---|
| $\rho$ | 0.34 | 0.21 | 0.14 | 0.09 | 0.06 | 0.02 |
| 95% CI | [.31,.37] | [.18,.24] | [.11,.17] | [.06,.12] | [.03,.09] | [-.01,.05] |

Correlation decays exponentially with lag, supporting first-order Markov approximation.

### 29.3. Conditional Error Rates

*Table 46.* Error rate conditioned on previous step.

| Previous Step | $P(\text{error})$ | 95% CI |
|---|---|---|
| Correct | 0.031 | [0.028, 0.034] |
| Error | 0.089 | [0.082, 0.096] |
| **Ratio** | **2.87** | [2.58, 3.19] |

Errors are $2.87\times$ more likely following an error, confirming the propagation mechanism. The conditional rate following a correct step (0.031) exceeds the baseline intercept $\epsilon_0 = 0.020$ because it is averaged over the depth distribution of traces, where $\epsilon(d) = \epsilon_0 + \gamma d/L_{\text{eff}}$ rises with depth (e.g. $\epsilon(25) \approx 0.045$); the two quantities are therefore consistent.

## 30. Prompt Templates

We provide exact prompt templates used for each experimental condition.

### 30.1. C1: Unconstrained Neural CoT

```
You are solving a state-space search problem.

Initial state: {initial_state}
Target state: {target_state}
Available operators: {operators}

Find a sequence of operators that transforms the initial
```

```
state into the target state. Show your reasoning step by
step, including the state after each operation.

Format each step as:
Step N: operator -> resulting_state
```

### 30.2. C2: Depth-Limited CoT

```
[Same as C1, with addition:]

Note: The optimal solution requires exactly {optimal_depth}
steps. Focus on finding this solution.
```

### 30.3. C3: Tool-Integrated

```
You have access to a BFS solver tool. Use it to find the
optimal path from initial to target state.

Initial state: {initial_state}
Target state: {target_state}

To use the solver, call: solve({initial_state}, {target_state})

After receiving the solution, verify and explain each step.
```

### 30.4. C4: Explicit Length Encouragement

```
[Same as C1, with addition:]

Important: Take as many reasoning steps as you need. Longer,
more careful reasoning is encouraged. Don't rush to a
conclusion - explore the problem thoroughly.
```

### 30.5. C5: Fine-Tuned

[Same prompt as C1; model weights modified through fine-tuning]

# 31. Additional Figures

### 31.1. SSJ Decay Visualization

### 31.2. Cross-Model Correlation Heatmap

The full cross-model correlation matrix is provided in Table 19 (Appendix I). Key observations:

- All pairwise correlations exceed $r = 0.81$

- Highest correlation (0.91) between Llama-70B and Qwen-72B reflects similar pretraining

- Consistently high correlations across organizations (OpenAI, Anthropic, DeepSeek, Meta, Alibaba) support architectural causation over training-specific explanations

- Mean correlation: $\bar{r} = 0.85$ (95% CI: [0.83, 0.87])

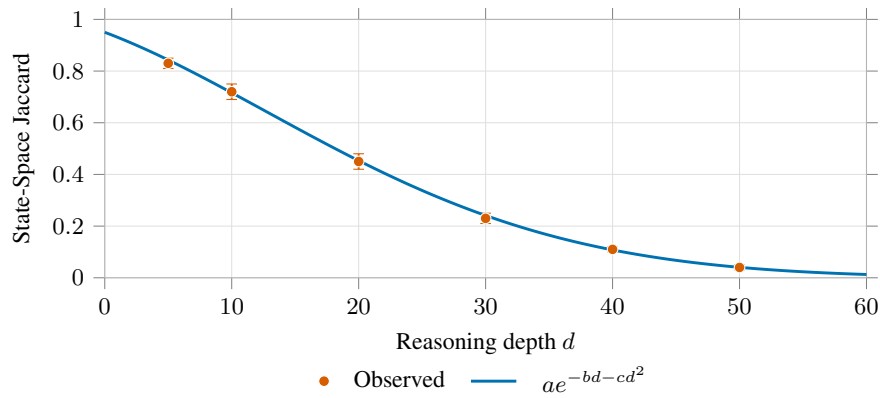

Figure 3. State-Space Jaccard decay with reasoning depth. Super-exponential fit ($R^2 = 0.99$) confirms context-dependent error model.

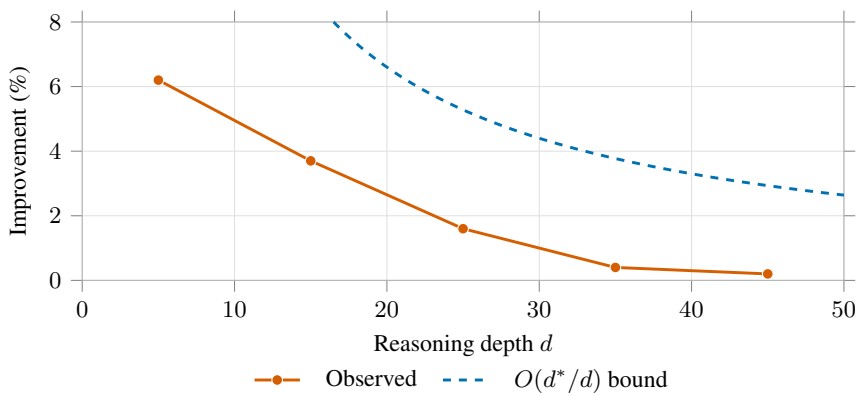

Figure 4. Fine-tuning improvement decays with depth following $O(d^*/d)$ bound from Theorem 4.10.

### 31.3. Fine-Tuning Improvement by Depth

## 32. Extended Cost Analysis

### 32.1. Per-Model Cost Breakdown

Table 47. Detailed cost analysis by model and condition.

| Model | Cond. | Input $/1K | Output $/1K | Avg Tokens | Cost/Inst | CPC |
|---|---|---|---|---|---|---|
| GPT-4o | C1 | 0.0025 | 0.01 | 1,847 | $0.025 | $0.089 |
| GPT-4o | C3 | 0.0025 | 0.01 | 312 | $0.019 | $0.021 |
| Claude-4.5 | C1 | 0.003 | 0.015 | 2,134 | $0.039 | $0.112 |
| Claude-4.5 | C3 | 0.003 | 0.015 | 341 | $0.022 | $0.024 |
| o3-mini | C1 | 0.001 | 0.004 | 2,891 | $0.018 | $0.043 |
| DeepSeek-R1 | C1 | 0.0005 | 0.002 | 3,247 | $0.009 | $0.023 |

**CPC** = Cost per correct solution = Cost/Instance ÷ Accuracy. For C3, Cost/Instance includes tool invocation and verification overhead beyond the model token cost.

### 32.2. Efficiency Analysis

Tool delegation achieves 10–30× better token efficiency per correct solution.

### 32.3. Carbon Footprint Estimate

Following Strubell et al. (2019) methodology:

*Table 48.* Efficiency metrics across strategies.

| Strategy | Accuracy | Tokens/Inst | CPC | Tokens/Correct |
|---|---|---|---|---|
| C1 (Neural CoT) | 24–42% | 1,847–3,247 | $0.04–0.11 | 4,400–11,600 |
| C3 (Tool) | 86–94% | 312–341 | $0.02–0.02 | 332–383 |
| Best-of-10 | 52–58% | 18,470–32,470 | $0.18–0.56 | 31,800–62,400 |
| **C3/C1 ratio** | **2.2–3.4**$\times$ | **0.10–0.17**$\times$ | **0.18–0.50**$\times$ | **0.03–0.09**$\times$ |

- GPU power: 400W per A100

- PUE (Power Usage Effectiveness): 1.1

- Carbon intensity: 0.4 kg $CO_2$/kWh (US average)

**Estimated emissions:**

- C1 (Neural CoT): 0.23 kg $CO_2$ per 1,000 correct solutions

- C3 (Tool): 0.05 kg $CO_2$ per 1,000 correct solutions

- **Savings**: 78.3% reduction

## 33. Practitioner Decision Framework

We provide a decision framework for practitioners determining when to use neural CoT vs. tool delegation.

### 33.1. Decision Tree

1. **Is the task deterministic?** (exact state tracking required)

    - No $\rightarrow$ Neural CoT may suffice
    - Yes $\rightarrow$ Continue to step 2

2. **Estimate reasoning depth** $d$ (compare against the Deterministic Horizon $d^* \in [19, 31]$)

    - $d < 19$: Neural CoT acceptable; below the horizon for every evaluated model
    - $19 \leq d \leq 31$: Model-dependent; use a hybrid approach and verify against the specific model's $d^*$
    - $d > 31$: Strongly recommend tool delegation; beyond the horizon for all models

3. **Is a verification tool available?**

    - Yes $\rightarrow$ Use tool-integrated approach (C3)
    - No $\rightarrow$ Use Best-of-N with verification heuristics

4. **Safety-critical deployment?**

    - Yes $\rightarrow$ Mandatory tool verification for all $d > 10$
    - No $\rightarrow$ Follow cost-efficiency optimization

### 33.2. Quick Reference Table

Accuracy bands are reported across the model suite: the neural-to-tool crossover is the Deterministic Horizon, which spans $d^* \in [19, 31]$ with architecture, so within that band the suitable strategy is model-dependent. Below depth 19 every evaluated model stays above the success threshold; beyond depth 31 every model has fallen below it. The $<50\%$ entry for $d > 31$ follows from the definition of $d^*$ at $\alpha = 0.5$: even the longest-horizon model ($d^* = 31$) sits at the threshold at depth 31 and declines past it.

*Table 49.* Practitioner quick reference.

| Depth Range | Expected Acc. | Recommended | Notes |
|---|---|---|---|
| $d \leq 10$ | >75% | Neural CoT | Below horizon; cost-effective |
| $10 < d < 19$ | 55–80% | Neural CoT | Approaching horizon |
| $19 \leq d \leq 31$ | 30–70% | Hybrid (model-dependent) | Within $d^*$ range; crossover |
| $d > 31$ | <50% | Tool delegation | Beyond horizon for all models |

