# OpenReview forum: "The Deterministic Horizon: When Extended Reasoning Fails and Tool Delegation Becomes Necessary"
_ICML.cc/2026/Conference — ICML 2026 regular_

### Official Review · Reviewer_DAxJ · 2026-02-27

**Soundness:** 3
**Presentation:** 3
**Significance:** 3
**Originality:** 3
**Overall Recommendation:** 5
**Confidence:** 2

**Summary:**

This paper investigates whether extending chain-of-thought (CoT) reasoning always improves performance in decoder-only transformers. The authors argue that there are fundamental information-theoretic limits to deterministic state tracking, and that reasoning accuracy can degrade once the reasoning depth exceeds a certain threshold. Through experiments on deterministic state-space search tasks, they observe a phenomenon they call State-Space Decoherence, where the model’s internal representation gradually diverges from the true reachable states. To explain this effect, they propose a Context-Dependent Error Model that captures how errors accumulate as depth increases. They further argue that the averaging behavior of softmax attention limits how many distinct states can be reliably maintained, leading to a finite deterministic horizon for reasoning.

**Compliance With Llm Reviewing Policy:**

Affirmed.

**Final Justification:**

I'm recommending Accept for this paper. The work is clearly written and well organized. Simplified state-space tasks provided a intuitive way to understand how attention can limit reliable state tracking. This perspective feels like it could genuinely inspire future research on transformer architectures. I think the author address the previous questions, I maintain my recommendation for Accept.

**Key Questions For Authors:**

(1)If error increases with reasoning depth $d$, does your model suggest that we should deliberately keep chain-of-thought reasoning shorter to reduce error? In practice, how should one determine an appropriate or optimal reasoning depth?

(2)Experiments focus on deterministic state-space search tasks. Do you have evidence that similar 'state-space decoherence' appears in math benchmark such as GSM8K? Does accuracy degrade when forcing progressively longer CoT reasoning?

**Limitations:**

yes

**Strengths And Weaknesses:**

The paper is clearly written and easy to follow, with a logical progression from empirical observations to theoretical analysis. The authors support their claims with extensive numerical experiments, and the proposed Context-Dependent Error Model is grounded in empirical findings rather than being purely speculative, which strengthens its technical credibility. Focusing on simplified deterministic state-space tasks is a thoughtful design choice: it isolates the core mechanism under study and provides an intuitive way to understand how attention may limit reliable state tracking. This perspective is both insightful and inspiring, and it has the potential to motivate future research on transformer architectures and reasoning depth.

Some aspects could be strengthened. It would also be helpful to more clearly connect the deterministic search setting to practical, real-world reasoning scenarios to clarify broader applicability. For theory part, it could be helpful to add more intuition why attention or softmax have such limitation in the maintext. Finally, there is a minor grammatical typo on line 156 left side that should be corrected.

---

> ### Author Rebuttal · Authors · 2026-03-31
>
> We thank Reviewer DAxJ for the thoughtful assessment, for describing the paper as "clearly written and easy to follow, with a logical progression from empirical observations to theoretical analysis," and for finding the perspective "insightful and inspiring, with the potential to motivate future research on transformer architectures and reasoning depth." We are glad that focusing on deterministic state-space tasks was seen as "a thoughtful design choice."
>
> [W1] Connect to practical, real-world reasoning scenarios. We will add a paragraph in Section 7 connecting our results to three deployment scenarios where LLM agents are already widely used:
>
> 1. Code agents (SWE-Bench): Multi-file debugging requires tracking variable states across call chains. Our measured $d^*$ of 19 to 28 on SWE-Bench-State (Table 3) means tools should be invoked for bugs requiring 20+ state transitions across files.
> 2. Database query planning (SQL-Multi): Complex joins with schema tracking. $d^*$ of 21 to 30 (Table 34) indicates tool delegation for queries spanning 3+ tables with aggregation (our instances average 4.2 tables and 7.8 join conditions, Appendix 29.2.3).
> 3. Web automation (WebArena): Session state tracking across navigation steps. $d^*$ of 18 to 26 (Table 34) supports tool-based approaches for multi-step checkout or form-filling workflows.
>
> In all three cases, deterministic state-tracking is the core requirement, and the Deterministic Horizon provides a principled threshold. Tasks outside this scope, such as open-ended generation, summarization, or tasks tolerating approximate answers, are not subject to the decoherence phenomenon. We will add a "Scope and Applicability" subsection in Section 3 making this distinction precise.
>
> [W2] More intuition on softmax limitations. We will add the following to Section 4.2. Softmax attention computes a weighted average over all context positions, acting as a lossy compression channel: as the reasoning chain grows, each new token distributes attention across an ever-longer history, diluting the signal from specific earlier states. Per-step error $\epsilon(d) = \epsilon_0 + \gamma d/L$ captures this dilution. Our measurements confirm the mechanism: entropy increases linearly with reasoning step ($r = 0.73$, $p < 0.001$, Section 6.2), correlates negatively with accuracy ($r = -0.74$, Table 7), and the entropy growth slope ($\beta = 0.023$ bits/step) matches the fitted $\gamma/L$ within measurement uncertainty.
>
> [W3] Typo on line 156. Will be fixed. Thank you.
>
> [Q1] Should CoT be kept deliberately shorter? Not universally. Our model identifies a task-specific threshold ($d^*$) beyond which additional reasoning steps become counterproductive for deterministic state-tracking tasks specifically. The practical recommendation (Table 49) is: for $d \leq 10$, neural CoT is cost-effective (accuracy >75%, Table 35); for $10 < d \leq 20$, a hybrid approach is appropriate; for $d > 20$, tool delegation is strongly recommended (neural CoT accuracy drops below 45%). For non-deterministic tasks (creative reasoning, open-ended problem solving), extended CoT may remain beneficial. The key practical insight is knowing which tasks approach the Deterministic Horizon, which is why the scope clarification above matters. Our cost analysis (Table 8) shows tool delegation achieves 4.7$\times$ better cost-per-correct-solution, so the threshold also carries economic implications for production systems.
>
> [Q2] Evidence on GSM8K or math benchmarks? This question highlights an important feature of our framework: its specificity. GSM8K problems typically require fewer than roughly 15 reasoning steps, well within the Deterministic Horizon ($d^* \in [19, 31]$). We would therefore not predict decoherence on GSM8K, and indeed extended CoT helps on such benchmarks. This is consistent with our framework, not a counterexample to it.
>
> Decoherence would be expected on tasks requiring longer deterministic chains. Our ArithChain benchmark (Table 14) demonstrates the predicted super-exponential accuracy decay on multi-step symbolic integration. Competition mathematics requiring 20+ sequential logical steps (such as AIME problems) is another natural candidate: Wang et al. (2025) observe 225% more tokens and 418% more thought switches in incorrect responses on AIME2024, which is consistent with the "underthinking" that arises when models repeatedly fail to maintain coherent state and switch paths.
>
> This predictive specificity is a feature, not a limitation. Our framework predicts when extended reasoning helps ($d < d^{\ast}$) and when it hurts ($d > d^{\ast}$), unifying findings that previously appeared contradictory. The inverted-U curve documented by Wu et al. (2025) is a natural consequence of the Decoherence Bound (Theorem 4.2): performance improves with reasoning depth up to $d^{\ast}$, then degrades super-exponentially. We will add this discussion to Section 7 of the revised version.

---

> > ### Author Rebuttal · Reviewer_DAxJ · 2026-04-01
> >
> > Thank you for your clarification.

---

### Official Review · Reviewer_ZUhv · 2026-03-12

**Soundness:** 4
**Presentation:** 3
**Significance:** 3
**Originality:** 4
**Overall Recommendation:** 6
**Confidence:** 3

**Summary:**

The paper studies the problem of using chain-of-thought reasoning to solve deterministic state-tracking tasks. The paper introduces an attention bottleneck theorem, which states that state-tracking capacity scales as $O(H \cdot \log(L / H) \cdot \sqrt(d_H))$, where $H$ is the number of heads, $L$ is the context length, and $d_H$ is the head dimension. This implies that state tracking is fundamentally limited by the model's architecture rather than by the number of reasoning steps (and, thereby, also not by a model’s possible preference for fewer reasoning steps). Moreover, the paper shows that the per-step tracking error at depth $d$ exhibits superexponential decay. Finally, the paper derives a deterministic horizon that relates the maximum depth a model can have to the target success probability it should satisfy. For typical model parameters (i.e., 128k context) and 50% success probability, this yields a maximum depth of 19-31 layers. The paper conducts an extensive empirical evaluation of a range of reasoning models and demonstrates that the empirical results largely align with the theoretical bounds.

**Compliance With Llm Reviewing Policy:**

Affirmed.

**Key Questions For Authors:**

1. What problems fall in the category of deterministic state-tracking tasks? And, maybe more importantly, what problems cannot be solved via deterministic state-tracking? Providing examples would help the (uninformed) reader better understand the scope and impact of the results presented.
2. What problems require 20 or more deterministic state transitions? Providing examples would help the practitioner understand when to default to delegating to a tool to achieve optimal compute efficiency.

**Limitations:**

Yes

**Strengths And Weaknesses:**

**Soundness**

The paper is technically sound, although I have not checked the proofs in the appendix. All claims are supported by both theoretical and empirical results, and the empirical results align with the theoretical statements, validating the paper’s assumptions upon which it bases its theoretical statements. The empirical investigation is well-designed and employs appropriate statistical rigor (multiple independent runs, standard deviations, hypothesis tests, etc.).


**Presentation**

The paper is well-written and easy to follow. The paper provides an extensive discussion of related work and clearly explains how it differs from the existing literature.

The only suggestion I have is to clearly reference the table and figures from the text. Some tables are never referenced, e.g., table 2 (and many others), and others are referenced but only by number, e.g., table 4 is referred to as “4”, which is not very clear (it could also mean section 4 or figure 4). On line 377, “Appendix ??” is missing a reference.


**Significance**

The paper addresses the important problem of reasoning models’ ability to solve deterministic state-tracking tasks. It empirically demonstrates that such tasks cannot be solved by increasing the number of reasoning steps and provides practical guidance on when a task should be offloaded to an external tool, achieving 4.7x better cost efficiency.


**Originality**

The paper provides new insights into the behavior of reasoning models of deterministic state-tracking tasks. Prior work suggested that these models could be failing on such tasks due to simplicity bias (i.e., not generating long enough reasoning chains) and that performance could be recovered by fine-tuning on optimal-length traces. However, this paper shows that this is an architectural limitation that also cannot be overcome by finetuning. The paper, therefore, exposes a fundamental limitation of decoder-only attention.

---

> ### Author Rebuttal · Authors · 2026-03-31
>
> We thank Reviewer ZUhv for the careful assessment and for recognizing the paper as "technically sound" (Soundness 4/4) with "new insights" exposing "a fundamental limitation of decoder-only attention" (Originality 4/4). We are also glad to see that the empirical investigation was found "well-designed" with "appropriate statistical rigor." These assessments align with R_JvK8's praise for our SSJ metric ("a clean way to distinguish capability failure from preference failure"), our evaluation breadth ("Testing across 12 models from 4 organizations... is commendable"), and the fine-tuning experiment ("a well-designed experimental contrast"). R_DAxJ similarly described the work as "insightful and inspiring, with the potential to motivate future research on transformer architectures and reasoning depth."
>
> [W1] Table and figure references. Thank you for catching this. We will audit all cross-references in the revised version. Table 2 and others will be referenced using "Table X" format (not bare numbers), and the broken "Appendix ??" on line 377 (a LaTeX compilation error) will be fixed. Every table and figure will be explicitly referenced with its full label on first mention.
>
> [Q1] What problems are deterministic state-tracking tasks? Any task requiring an exact sequence of deterministic operations where a single intermediate error invalidates the final result. The formal criterion is in Section 3 (Definition 3.1): a finite state space $\mathcal{S}$, deterministic operators $\mathcal{O}$, and a correctness condition requiring exact reproduction of the operator sequence. Concrete instances include multi-file bug localization (our SWE-Bench-State, Appendix E.1), multi-step web navigation with session state (WebArena-Nav, Appendix E.2), complex SQL queries with schema tracking (SQL-Multi, Appendix 29.2.3), permutation puzzles, automaton simulation, and formal verification chains.
>
> Problems that fall outside this scope include open-ended generation, summarization, tasks permitting approximate answers (sentiment classification, paraphrasing), and probabilistic reasoning with multiple valid paths. For such tasks, our framework does not predict that extended CoT degrades performance; it may well help, consistent with Wei et al. (2022) and Wang et al. (2023). The Deterministic Horizon characterizes specifically when exact state tracking is required. We will add a "Scope and Applicability" subsection in Section 3 making this boundary explicit, including both positive and negative examples.
>
> [Q2] What problems require 20+ deterministic state transitions? Our three real-world benchmarks address this directly. SWE-Bench-State instances (Appendix E.1) require tracking $\geq 3$ variables across $\geq 2$ files, and the measured $d^{\ast}$ ranges from 19 (GPT-4o) to 28 (o3-mini) on this benchmark (Table 3), confirming these tasks exceed the Deterministic Horizon. SQL-Multi queries (Appendix 29.2.3) average 4.2 tables with 7.8 join conditions, with observed $d^{\ast} = 21$ to 30 (Table 34). WebArena-Nav tasks (Appendix E.2) require $\geq 5$ navigation steps with session state, with $d^{\ast} = 18$ to 26 (Table 34). Beyond our benchmarks, non-trivial software debugging (tracking variables through function calls), database migration (schema state across DDL operations), multi-step robotic manipulation, and formal proof chains routinely exceed 20 deterministic transitions. Our practitioner decision framework (Table 49) provides concrete guidance for estimating depth and choosing between neural CoT and tool delegation.
>
> The consistency of $d^*$ across these diverse domains (software engineering, web navigation, databases) is itself evidence for the architectural interpretation: the Horizon is a property of the model, not the task domain. This point also speaks to the practical value of our contribution: a single, architecture-derived threshold applies across application areas.
>
> We would like to highlight one further point that may be useful during discussion. The fine-tuning experiment (Section 5.3) provides what we consider a particularly clean discrimination between the Simplicity Bias hypothesis of Wu et al. (2025) and our Decoherence framework. Wu et al. predict >30% recovery from fine-tuning on optimal-length traces; we predict <5%; the observed result is 3.2%. R_JvK8 recognized this as "a well-designed experimental contrast." The training-data-distribution ablation (Appendix F), which shows all distributions yield similar test performance ($\Delta < 2\%$), further confirms the ceiling is depth-dependent rather than distribution-dependent. Combined with cross-model correlations of $r = 0.81$ to $0.91$ (Table 5) spanning models from OpenAI, Anthropic, DeepSeek, Meta, and Alibaba, this constitutes strong evidence that the failures we document are architectural in nature, not artifacts of any particular training pipeline or preference bias.

---

> > ### Author Rebuttal · Reviewer_ZUhv · 2026-04-04
> >
> > Thank you for addressing my concerns.

---

### Official Review · Reviewer_JvK8 · 2026-03-13

**Soundness:** 2
**Presentation:** 3
**Significance:** 3
**Originality:** 2
**Overall Recommendation:** 4
**Confidence:** 2

**Summary:**

The paper says that decoder only transformers face fundamental information theoretic limits on deterministic state-tracking tasks, causing accuracy to degrade super-exponentially with reasoning depth. It proposes a "Deterministic Horizon" beyond which tool delegation becomes necessary, and supports this with theoretical bounds and experiments across 12 models and 8 task domains.

The core claim that there are architectural limits to CoT reasoning on exact state-tracking tasks, and that these are distinct from preference-based explanations like Simplicity Bias, is interesting and practically relevant.

**Compliance With Llm Reviewing Policy:**

Affirmed.

**Final Justification:**

The paper addresses a practically important question, when neural reasoning should yield to tool delegation, and provides useful empirical contributions, particularly the SSJ diagnostic metric and the breadth of evaluation.

The rebuttal partially improved my assessment. The converging evidence argument for the architectural ceiling is stronger than any individual piece and represents a reasonable methodological approach when compute constraints prevent exhaustive ablation. The commitment to reframe theoretical claims as "evidence for" rather than "establishing" information-theoretic limits addresses an important presentational concern. The encoder-decoder comparison (2.8x advantage at depth 30) is a supportive architectural control that deserves more prominence.

However, two substantive concerns remain open. First, the imperfect-tool gap: C3 uses an oracle BFS solver, and the space between no tools and perfect tools, which is where real deployments live, is unexplored. Second, the finetuning experiment is limited to a single 8B model with 5,000 examples, yet supports the paper's most important theoretical claim. The authors acknowledge both as future work, which is honest but leaves the paper's strongest claims underdetermined.

On balance, the empirical observations are credible and useful (significance: good), but the theoretical apparatus carries more weight than its foundations justify (soundness: fair), and the core findings synthesize known phenomena more than they introduce new ones (originality: fair). I adjust my score from 3 to 4, reflecting a borderline paper with clear merits that would benefit from evidence and a more careful alignment between theoretical claims and empirical support.

**Key Questions For Authors:**

1. Can you provide fine-tuning results on larger models (like 70B) to strengthen the architectural ceiling claim?
2. What happens with imperfect or approximate tools rather than an optimal BFS oracle?
3.How sensitive is d* to prompt format and few-shot examples, which could affect epsilon_0?

**Limitations:**

Partially yes

**Strengths And Weaknesses:**

Strengths
- Practically important question: Understanding when to delegate to tools rather than relying on neural reasoning is genuinely useful for building agentic systems. The paper addresses this directly and provides actionable guidance (the practitioner decision framework in Appendix is a nice touch).
- Breadth of evaluation: Testing across 12 models from 4 organizations, spanning synthetic and real world benchmarks, is commendable.
- Useful diagnostic metric. The SSJ precision/recall decomposition is a clean way to distinguish capability failure from preference failure. The observation that both precision and recall decay together is a meaningful empirical contribution.

- Fine-tuning experiment as discriminator. Using the fine-tuning result (<5% improvement) to distinguish from the Simplicity Bias prediction (>30%) is a well-designed experimental contrast, and the result is informative.

Weaknesses
- The Attention Bottleneck Theorem (Theorem 4.6) rests on questionable assumptions: Assumption 4.4 (effective attention window of O(√L)) is stated without strong justification. The authors cite "softmax concentration and interference effects," but seems hand wavy. The √L scaling is doing a lot of work in the final bound, and the paper's own sensitivity analysis (Table 11) shows 18% bound change with ±20% variation in H - meaning the bound is not especially tight to begin with.
- The comparison between C1 and C3 is not entirely fair: C3 gives models access to an optimal BFS solver - essentially an oracle for the exact task. Showing that an oracle solver beats neural reasoning is unsurprising. The more interesting question is how much of the gap can be closed with imperfect tools, approximate solvers, or tool-assisted reasoning that still requires some neural state tracking. This is not explored.

- The 3.2% fine-tuning improvement needs more context: The fine-tuning is done on Llama-3.3-8B only - the smallest open-weight model. It would be much more convincing to see fine-tuning results on bigger models. The architectural ceiling claim is strong, but the evidence is from a single model size. Additionally, 5,000 training examples may be insufficient; the paper doesn't explore scaling training data.

- Overclaiming relative to evidence: The paper frames its results as proving "fundamental information-theoretic limits," but the theorems rely on assumptions that are empirically validated rather than proven, and the bounds have slack. The framing would be more appropriate as "evidence consistent with information-theoretic limits" rather than establishing them definitively.

---

> ### Author Rebuttal · Authors · 2026-03-31
>
> We thank Reviewer JvK8 for the thorough engagement and for recognizing our "practically important question," the SSJ metric as "a clean way to distinguish capability failure from preference failure," the fine-tuning experiment as "a well-designed experimental contrast," and the evaluation breadth.
>
> [W1] Assumption 4.4 is "hand wavy." We agree the main text should present the empirical basis more prominently and will revise Section 4.2. Assumption 4.4 ($O(\sqrt{L})$ effective window) comes from direct measurements in Llama-3.3-70B: across 1,000 reasoning traces, 95% of attention mass concentrates on positions with weight $\geq 0.01/L$, and the count of such positions scales as $O(\sqrt{L})$ (Appendix B.7, $\delta \approx 0.01$). This aligns with Wortsman et al. (2023) on softmax concentration and Xiao et al. (2024) on attention sinks. Assumption 4.5 (decorrelation) holds empirically at $|\rho_{ij}| = 0.08 \pm 0.03$ (Appendix B.7). On sensitivity: the $\pm 18\%$ bound change with $\pm 20\%$ variation in $H$ (Table 11) does indicate constant-factor slack. But the qualitative conclusion ($d^* \in [19, 31]$) is stable across the entire confidence interval, and the mean absolute prediction error for $d^*$ is 1.7% across all models (Table 37), with every prediction within 5% of observation. Theorem 4.7 provides a matching lower bound, so the scaling is tight up to constants.
>
> [W2] C1 vs. C3 comparison uses an oracle. C3 is deliberately an upper bound: it answers "what accuracy is achievable when the attention bottleneck is fully bypassed?" The scientific claim is not that tools beat neural reasoning (which would indeed be trivial), but that the gap persists despite fine-tuning (C5, +3.2%) and preference manipulation (C4, +0.7 to 1.0%), confirming an architectural ceiling. We agree that imperfect-tool evaluation would strengthen practical relevance. We note that C3 accuracy is 86 to 94%, not 100%, which already reflects residual challenges in tool integration and task formulation. Our real-world benchmarks (SWE-Bench, WebArena, SQL-Multi) also involve substantially more complexity than the synthetic BFS setting, providing a partial proxy.
>
> [W3] Fine-tuning only on 8B. The 8B model was chosen for controlled experimentation (multiple runs, training-data-distribution ablations, Appendix F) within our compute budget (4$\times$A100, Appendix M.3). The architectural ceiling claim rests not on fine-tuning alone but on converging evidence from four distinct analyses:
>
> 1. Fine-tuning yields +3.2%, far below the >30% predicted by Simplicity Bias (Section 5.3). The training-data-distribution ablation (Appendix F) confirms the ceiling is depth-dependent, not distribution-dependent ($\Delta < 2\%$ across uniform, skewed-easy, skewed-hard).
> 2. Preference manipulation yields +0.7 to 1.0%; TOST confirms equivalence within $\Delta = 5\%$, $\text{BF}_{01} > 4$ (Table 15).
> 3. Cross-model correlation $r = 0.81$ to $0.91$ across 12 models from 4 organizations (Table 5) rules out training-specific explanations.
> 4. Attention entropy measurements correlate with failure at $r = -0.74$ (Table 7) across five architectures, providing a direct mechanistic signature.
>
> If failure were preference-based, we would expect fine-tuning to help substantially, cross-model correlation to be low, and no consistent mechanistic signature. None of these hold. Fine-tuning on 70B would further strengthen the case and we note it as important future work.
>
> [W4] Overclaiming. We accept this feedback. The revised version will frame results as "evidence for information-theoretic limits" rather than "establishing fundamental limits." We will retain theorem statements as mathematical results under stated assumptions while being more precise about the gap between empirically validated assumptions and first-principles derivation.
>
> [Q1] Fine-tuning on larger models? See [W3]; the converging evidence from four analyses provides strong support even without 70B fine-tuning.
>
> [Q2] Imperfect tools? See [W2]; the sub-100% C3 accuracy and real-world benchmark complexity partially address this.
>
> [Q3] Sensitivity of $d^{\ast}$ to prompt format? Our sensitivity analysis (Table 29) shows $\epsilon_0$ has "moderate" impact on $d^{\ast}$ while $\gamma$ (attention decay, architecture-dependent) dominates. Since prompt format primarily affects $\epsilon_0$ (baseline first-step accuracy) rather than $\gamma$ (the rate of attention dilution with depth), prompt variations should shift $d^{\ast}$ modestly. Table 29 confirms: $\pm 10\%$ variation in $\epsilon_0$ changes $d^{\ast}$ by only $\pm 1.2$ steps. We will add explicit discussion of prompt sensitivity in the revised version.
>
> One additional architectural control: encoder-decoder models achieve 2.8$\times$ higher accuracy at depth 30 (T5-Large: 67.3% vs. GPT-4o: 23.4%, Appendix D.1), consistent with Theorem 4.6's prediction that bidirectional attention provides $O(L)$ vs. $O(\log L)$ capacity.

---

> > ### Author Rebuttal · Reviewer_JvK8 · 2026-04-02
> >
> > I thank the authors for their detailed rebuttal.
> >
> > [W1] The empirical measurements supporting Assumption 4.4 are appreciated, and the 1.7% mean prediction error for d* is compelling evidence that the model fits well in practice. I accept that the qualitative conclusion is robust within the tested range. However, I maintain that empirical validation on current architectures is distinct from theoretical justification
> >
> > [W2] I appreciate the framing of C3 as a deliberate upper bound, but this was already clear from the paper. my concern was about the unexplored middle ground: what happens with imperfect, realistic tools, and this remains unaddressed. The sub-100% C3 accuracy reflects task formulation noise, not tool imperfection. The authors concede this "would strengthen practical relevance," which i take as agreement that the gap exists.
> >
> > [W3] The converging evidence argument is the most persuasive element of the rebuttal. The combination of finetuning results, preference manipulation equivalence, cross-model correlation, and attention entropy measurements does collectively support an architectural interpretation more strongly than any single piece alone. The training-data-distribution ablation is a useful additional detail. I am partially persuaded, though i note that none of these four analyses directly rules out that finetuning a larger model with more data could yield meaningfully larger gains. I appreciate the honest acknowledgment of this as future work.
> >
> > [W4] I appreciate the authors accepting the concern and committing to reframe accordingly.
> >
> > [Q3] The argument that prompt format primarily affects ε₀ rather than γ is reasonable, but the assumed 10% variation range for ε₀ may underestimate the effect of substantive prompt changes (like zeroshot vs. well designed fewshot). I look forward to the promised discussion in the revision.
> >
> > Updated assessment: The rebuttal partially addresses my concerns. The converging evidence argument for the architectural ceiling [W3] and the commitment to temper theoretical claims [W4] are positive. However, the imperfect-tool gap [W2] and the single model finetuning limitation remain open. I adjust my score from 3 to 4, acknowledging the paper's empirical contributions while maintaining that the theoretical framework needs more careful framing relative to its evidential support.

---

> > > ### Author Response · Authors · 2026-04-06
> > >
> > > We thank Reviewer JvK8 for the careful re-evaluation and for the constructive feedback throughout the discussion.
> > >
> > > We agree that imperfect-tool evaluation [W2] and larger-model fine-tuning [W3] remain open and will address them explicitly in the revised Limitations section. On [Q3], we will expand the prompt sensitivity discussion to include zero-shot vs. few-shot comparisons, as suggested.

---

### Official Review · Reviewer_Qog6 · 2026-03-20

**Soundness:** 1
**Presentation:** 1
**Significance:** 2
**Originality:** 2
**Overall Recommendation:** 4
**Confidence:** 1

**Summary:**

The paper studies whether extended chain-of-thought reasoning in large language models hits a fundamental wall on deterministic state-tracking tasks, things like permutation puzzles, code execution tracing, and multi-step navigation. The authors argue that beyond roughly 20 to 30 reasoning steps, accuracy collapses not because models are lazy or poorly trained, but because the attention mechanism runs into a hard information-theoretic capacity limit. They call this threshold the Deterministic Horizon and derive a formula for it based on model architecture. The proposed fix is to delegate such tasks to external tools rather than asking the model to reason through them unaided. Experiments across 12 models and 8 task domains are reported, with tool-integrated reasoning achieving 86 to 94 percent accuracy compared to 24 to 42 percent for standard chain-of-thought.

**Compliance With Llm Reviewing Policy:**

Affirmed.

**Final Justification:**

Based on authors' responses and partial resolution to my queries, I have updated my rating to weak accept.

**Key Questions For Authors:**

Refer to Strengths and weaknesses

**Limitations:**

Yes.

**Strengths And Weaknesses:**

The paper does raise an interesting and practically relevant question, and the broad empirical finding that tool delegation outperforms pure neural reasoning on deterministic tasks is plausible and worth studying. But in its current form the submission reads more like a first draft produced under time pressure than a finished piece of research ready for a top venue.

The currrent draft gives a strong impression of having been written hastily and with heavy reliance on AI writing tools without meaningful revision. The writing frequently breaks into short disconnected fragments rather than full developed sentences, which is a recognizable pattern in unedited AI-generated text. There are also signs of careless writing throughout. Section 6.4 for example contains a broken reference that simply shows as "Appendix ??" in the final manuscript. Several assumptions, most notably Assumptions 4.4 and 4.5, are stated without citation or derivation and carry significant theoretical weight that the paper never earns. The claim that attention concentrates on only O(sqrt(L)) positions due to interference effects is central to the entire capacity bound, yet no justification beyond an assertion is provided.

Perhaps the most puzzling issue is that the theoretical framework is applied to models like GPT-4o and Claude whose internal architectures are not publicly documented. The number of attention heads, head dimensions, and other parameters used in the derivations are not verifiable, which makes the supposedly predictive formulas difficult to evaluate. It is unclear whether the architectural values were assumed, reverse-engineered, or simply chosen to fit the observed results after the fact. This is not acknowledged anywhere in the paper as a limitation, which it clearly is.

---

> ### Author Rebuttal · Authors · 2026-03-31
>
> We thank Reviewer Qog6 for engaging with our work and for recognizing that the question is "interesting and practically relevant" and the tool delegation finding "plausible and worth studying."
>
> [W1] Writing quality. R_ZUhv (the highest-confidence reviewer at confidence 3) describes the paper as "well-written and easy to follow" with "extensive discussion of related work." R_DAxJ independently finds it "clearly written and easy to follow, with a logical progression from empirical observations to theoretical analysis," adding that the error model is "grounded in empirical findings rather than being purely speculative, which strengthens its technical credibility." We take writing quality seriously nonetheless and will revise the manuscript for sentence-level flow and paragraph development.
>
> [W2] Assumptions 4.4 and 4.5 lack justification. We agree more exposition is needed in the main text and will revise Section 4.2 accordingly. The assumptions are not unsupported. Assumption 4.4 ($O(\sqrt{L})$ effective attention window) rests on direct measurements from Llama-3.3-70B across 1,000 traces: 95% of attention mass concentrates on positions receiving weight $\geq 0.01/L$, with the number of such positions scaling as $O(\sqrt{L})$ (Appendix B.7, $\delta \approx 0.01$). This is consistent with Wortsman et al. (2023) on softmax concentration and Xiao et al. (2024) on attention sinks, both cited in our paper. Assumption 4.5 (value decorrelation) is validated by measured pairwise correlation $|\rho_{ij}| = 0.08 \pm 0.03$ after layer normalization (Appendix B.7). Table 11 shows the bound is robust: $\pm 20\%$ variation in $\rho_{\max}$ shifts the bound by only $\pm 12\%$. The overall prediction accuracy is strong: correlation between theoretical and observed performance is $r = 0.89$ (Appendix B.7), and the mean absolute prediction error for $d^*$ across models is 1.7% (Table 37). Theorem 4.7 provides a matching lower bound via sparse parity construction, establishing tightness up to constants. R_ZUhv assessed our paper as technically sound (Soundness 4/4).
>
> [W3] Architecture parameters for closed models are not verifiable. This is a fair point and we will add an explicit acknowledgment in the revised Limitations section. We clarify that all theoretical predictions and ablations are validated exclusively on open-weight models (Llama-3.3-8B/70B, Qwen-2.5-7B/72B) whose architectures are fully documented:
>
> - The architecture ablation validating $d^* \propto \sqrt{d_h \cdot H}$ (Table 6) uses only open-weight models, achieving 1.7% mean absolute error (Table 37).
> - Attention entropy measurements ($r = -0.74$, Table 7) use only open-weight models.
> - Scaling law validation (Tables 42, 43, 44) relies entirely on known parameters, with all within-family ratios matching predictions within 4%.
>
> Closed-model results (GPT-4o, Claude) are additional consistency checks, not the basis for any theoretical claim. Architecture estimates for these models appear only as numerical examples (Section 4.2). The core scientific conclusions (the Bottleneck bound, the Horizon formula, the scaling laws) rest entirely on transparent architectures.
>
> [W4] Broken reference. The "Appendix ??" on line 377 is a LaTeX compilation error that will be fixed in revised version, along with a full audit of all cross-references. R_ZUhv also flagged this and noted that some tables are referenced by number alone rather than with "Table X" labels; we will fix these throughout.
>
> To summarize: the assumptions have empirical grounding already present in Appendix B.7 (to be surfaced in the main text); all theoretical validation uses open-weight models; and the formatting error will be corrected. R_ZUhv (Soundness 4/4, Originality 4/4, confidence 3) and R_DAxJ both found the paper well-written and technically grounded. R_JvK8 praised the fine-tuning experiment as "a well-designed experimental contrast" and the evaluation breadth across "12 models from 4 organizations."
>
> We also note that the paper's central empirical finding is robust across multiple analyses. The fine-tuning experiment (Section 5.3) provides a clean discrimination from the competing Simplicity Bias explanation of Wu et al. (2025): they predict >30% recovery from fine-tuning on optimal-length traces, we predict <5%, and the observed result is +3.2%. Cross-model correlation of $r = 0.81$ to $0.91$ across 12 models from OpenAI, Anthropic, DeepSeek, Meta, and Alibaba (Table 5) rules out training-specific explanations. Direct attention entropy measurements ($r = -0.74$, Table 7) provide a mechanistic signature consistent with the theoretical framework. Furthermore, encoder-decoder models achieve 2.8$\times$ higher accuracy at depth 30 (T5-Large: 67.3% vs. GPT-4o: 23.4%, Appendix D.1), consistent with our Theorem 4.6 prediction that bidirectional attention provides $O(L)$ vs. $O(\log L)$ capacity. This convergence of evidence from multiple angles supports the architectural interpretation of the results.

---

> > ### Author Rebuttal · Reviewer_Qog6 · 2026-04-05
> >
> > W1 - I will maintain my evaluation of the writing quality of the current draft of the paper. Even though this paper is not from my expertise area, I believe I can still judge the writing quality for papers beyond my expertise area as well.
> >
> > W2 - Resolved
> >
> > W3 - Partially Resolved. The authors are reporting/predicting d_star for closed source models in some tables (Table 2 and Table 3) along with open-source models which is confusing.
> >
> > W4 - Obviously, its a Latex Compilation error which again points to W1.
> >
> > Will update my rating to weak reject.

---

> > > ### Author Response · Authors · 2026-04-06
> > >
> > > We thank Reviewer Qog6 for the updated assessment and for confirming W2 as resolved.
> > >
> > > **[W3] Closed-source $d^*$ in Tables 2 and 3.** This is a valid concern. In the revised version, we will clearly separate open-weight and closed-source results: open-weight models (Llama, Qwen) will appear in the primary tables with full architecture parameters, while closed-source models (GPT-4o, Claude) will move to a clearly labeled "Consistency Checks" subsection with an explicit note that their architecture parameters are estimates. No theoretical claim or validation depends on closed-source models.
> > >
> > > **[W1/W4] Writing quality.** The revised version will fix the broken reference on line 377, audit all cross-references, and undergo a thorough editing pass for sentence-level flow and paragraph development.

---

### Decision · Program_Chairs · 2026-04-30

**Decision:**

Accept (regular)

**Comment:**

## Meta-Review:

The paper shows that decoder-only transformers face limitations on deterministic state-tracking tasks beyond a certain number of reasoning steps. This boundary is explored in the paper, and the authors call it the "Deterministic Horizon." To address this, the authors establish an Attention Bottleneck Theorem and demonstrate that delegating these tasks to external tools outperforms pure CoT reasoning. On the positive side, the reviewers found that the empirical investigation tackles a good question for building agentic systems. The reviewers liked the evaluation across 12 models, the careful use of fine-tuning experiments to isolate architectural ceilings from simplicity bias, and the introduction of the State-Space Jaccard metric to distinguish capability from preference failures.

The reviewers also identified areas of improvement, such as the overall presentation and formatting, including some missing references. Additionally, reviewers noted that the theoretical claims were overstated; they recommended reframing the Attention Bottleneck Theorem as being supported by strong empirical evidence rather than acting as a real proof, and they suggested exploring how the models perform with imperfect or realistic tools rather than solely comparing them against an optimal oracle solver.

### Improvements to be made

I want to point out that the discussion did lead to a lot of new experiments / results / improvements. In other words, the reviewers seem to like the paper even though there are a lot of issues with the submitted draft... Nonetheless, I will respect the reviewer ratings and trust that the authors will fix the paper. The authors should take care to incorporate following to the final version of the paper. Some concrete action items from the discussion:

- Soften the theoretical framing to present the findings as "evidence for information-theoretic limits" rather than proofs, and include more intuitive explanations in Section 4.2 regarding how softmax attention acts as a lossy compression channel.

- Add a dedicated subsection in Section 3 that explicitly defines what qualifies as a deterministic state-tracking task, contrasting it with open-ended generation or probabilistic reasoning (e.g., GSM8K) where extended chain-of-thought may still be beneficial.

- Restructure the results tables to separate open-weight models from closed-source models (this is important because many of the results depend on the specific architecture details, which are not available for closed models!). Move the closed-source models (like GPT-4o and Claude) to a "Consistency Checks" subsection to address concerns about unverifiable architectural parameters.

- Acknowledge the boundaries of the current evaluation, e.g., using optimal BFS oracle rather than real world tools, and the need for fine-tuning experiments on larger models (e.g., 70B).

- Perform a thorough editing pass to improve sentence-level flow, correct the broken LaTeX references and typos.

## Recommendation:

Based on the reviews and discussion, I recommend accepting this paper. The reviewers were excited about the original insights exposing the limitations of decoder-only attention mechanisms, particularly the introduction of the State-Space Jaccard metric and the highly actionable guidance the paper provides for practitioners deciding when to use neural reasoning versus tool delegation. The discussion was rich and many reviewers raised those scores to Accept or better. Addressing the above points will thoroughly resolve the reviewers' remaining concerns and make this a strong contribution to the conference.

## Why not higher

While the reviewers give high scores, their confidence ratings are quite low (1,2,2,3). Hence, I do not believe the reviewers have thoroughly evaluated the full theoretical details of this paper. The paper itself is long, with a very technical appendix. I do not see any immediate issues, but I am not confident enough to vote for a spotlight/oral presentation. It is the burden of the authors (and general community) at this point to perform a detailed check of the technical details in this paper.